# Nitric oxide acts as a cotransmitter in a subset of dopaminergic neurons to diversify memory dynamics

Yoshinori Aso[1]*, Robert P Ray[1], Xi Long[1], Daniel Bushey[1], Karol Cichewicz[2], Teri-TB Ngo[1], Brandi Sharp[1], Christina Christoforou[1], Amy Hu[1], Andrew L Lemire[1], Paul Tillberg[1], Jay Hirsh[2], Ashok Litwin-Kumar[3], Gerald M Rubin[1]*

[1]Janelia Research Campus, Howard Hughes Medical Institute, Ashburn, United States; [2]Department of Biology, University of Virginia, Charlottesville, United States; [3]Department of Neuroscience, Columbia University, New York, United States

**Abstract** Animals employ diverse learning rules and synaptic plasticity dynamics to record temporal and statistical information about the world. However, the molecular mechanisms underlying this diversity are poorly understood. The anatomically defined compartments of the insect mushroom body function as parallel units of associative learning, with different learning rates, memory decay dynamics and flexibility (Aso and Rubin, 2016). Here, we show that nitric oxide (NO) acts as a neurotransmitter in a subset of dopaminergic neurons in *Drosophila*. NO's effects develop more slowly than those of dopamine and depend on soluble guanylate cyclase in postsynaptic Kenyon cells. NO acts antagonistically to dopamine; it shortens memory retention and facilitates the rapid updating of memories. The interplay of NO and dopamine enables memories stored in local domains along Kenyon cell axons to be specialized for predicting the value of odors based only on recent events. Our results provide key mechanistic insights into how diverse memory dynamics are established in parallel memory systems.

*For correspondence:
asoy@janelia.hhmi.org (YA);
rubing@janelia.hhmi.org (GMR)

Competing interests: The authors declare that no competing interests exist.

## Introduction

An animal's survival in a dynamically changing world depends on storing distinct sensory information about their environment as well as the temporal and probabilistic relationship between those cues and punishment or reward. Thus, it is not surprising that multiple distributed neuronal circuits in the mammalian brain have been shown to process and store distinct facets of information acquired during learning (*White and McDonald, 2002*). Even a simple form of associative learning such as fear conditioning induces enduring changes, referred to as memory engrams, in circuits distributed across different brain areas (*Herry and Johansen, 2014*). Do these multiple engrams serve different mnemonic functions, what molecular and circuit mechanisms underlie these differences, and how are they integrated to control behavior? Localizing these distributed engrams, understanding what information is stored in each individual memory unit and how units interact to function as one network are important but highly challenging problems.

The *Drosophila* mushroom body (MB) provides a well-characterized and experimentally tractable system to study parallel memory circuits. Olfactory memory formation and retrieval in insects requires the MB (*de Belle and Heisenberg, 1994*; *Dubnau et al., 2001*; *Erber et al., 1980*; *Heisenberg, 2003*; *McGuire et al., 2001*). In associative olfactory learning, exposure to an odor paired with a reward or punishment results in formation of a positive- or negative-valence memory, respectively (*Quinn et al., 1974*; *Tempel et al., 1983*; *Tully and Quinn, 1985*). In the MB, sensory stimuli are represented by the sparse activity of ~2000 Kenyon cells (KCs). Each of 20 types of dopaminergic

neurons (DANs) innervates compartmental regions along the parallel axonal fibers of the KCs. Similarly, 22 types of mushroom body output neurons (MBONs) arborize their dendrites in specific axonal segments of the KCs; together, the arbors of the DANs and MBONs define the compartmental units of the MB (*Aso et al., 2014a*; *Mao and Davis, 2009*; *Tanaka et al., 2008*). Activation of individual MBONs can cause behavioral attraction or repulsion, depending on the compartment in which their dendrites arborize, and MBONs appear to use a population code to govern behavior (*Aso et al., 2014b*; *Owald et al., 2015*).

A large body of evidence indicates that these anatomically defined compartments of the MB are also the units of associative learning (*Aso et al., 2012*; *Aso et al., 2014b*; *Aso et al., 2010*; *Berry et al., 2018*; *Blum et al., 2009*; *Bouzaiane et al., 2015*; *Burke et al., 2012*; *Claridge-Chang et al., 2009*; *Huetteroth et al., 2015*; *Ichinose et al., 2015*; *Isabel et al., 2004*; *Krashes et al., 2009*; *Lin et al., 2014*; *Liu et al., 2012*; *Owald et al., 2015*; *Pai et al., 2013*; *Plaçais et al., 2013*; *Schwaerzel et al., 2003*; *Séjourné et al., 2011*; *Trannoy et al., 2011*; *Yamagata et al., 2015*; *Zars et al., 2000*). Despite the long history of behavioral genetics in fly learning and memory, many aspects of the signaling pathways governing plasticity—especially whether they differ between compartments—remain poorly understood. Nevertheless, dopaminergic neurons and signaling play a key role in all MB compartments, and flies can be trained to form associative memories by pairing the presentation of an odor with stimulation of a single dopaminergic neuron (*Aso et al., 2010*). Punishment or reward activates distinct sets of DANs that innervate specific compartments of the MB (*Das et al., 2014*; *Galili et al., 2014*; *Kirkhart and Scott, 2015*; *Liu et al., 2012*; *Mao and Davis, 2009*; *Riemensperger et al., 2005*; *Tomchik, 2013*). Activation of the DAN innervating an MB compartment induces enduring depression of KC-MBONs synapses in those specific KCs that were active in that compartment at the time of dopamine release (*Berry et al., 2018*; *Bouzaiane et al., 2015*; *Cohn et al., 2015*; *Hige et al., 2015*; *Owald et al., 2015*; *Séjourné et al., 2011*). Thus, which compartment receives dopamine during training appears to determine the valence of the memory, while which KCs were active during training determines the sensory specificity of the memory (*Figure 1A*).

Compartments operate with distinct learning rules. Selective activation of DANs innervating specific compartments has revealed that they can differ extensively in their rates of memory formation, decay dynamics, storage capacity, and flexibility to learn new associations (*Aso et al., 2012*; *Aso and Rubin, 2016*; *Huetteroth et al., 2015*; *Yamagata et al., 2015*). For instance, the dopaminergic neuron PAM-$\alpha$1 can induce a 24 hr memory with a single 1 min training session, whereas PPL1-$\alpha$3 requires 10 repetitions of the same training to induce a 24 hr memory. PPL1-$\gamma$1pedc (aka MB-MP1) can induce a robust short-lasting memory with a single 10 s training, but cannot induce long-term memories even after 10 repetitions of a 1 min training. PAM-$\alpha$1 can write a new memory without compromising an existing memory, whereas PPL1-$\gamma$1pedc extinguishes the existing memory when writing a new memory (*Aso and Rubin, 2016*). What molecular and cellular differences are responsible for the functional diversity of these compartments? Some differences might arise from differences among KC cell types (reviewed in *Keene and Waddell, 2007*; *McGuire et al., 2005*), but memory dynamics are different even between compartments that lie along the axon bundles of the same Kenyon cells (for example, $\alpha$1 and $\alpha$3). In this paper, we show that differences in memory dynamics between MB compartments can arise from the deployment of distinct cotransmitters by the DAN cell types that innervate them.

## Results

### Dopaminergic neurons can induce memories without dopamine, but with inverted valence

DANs release diverse cotransmitters in the mammalian brain (*Maher and Westbrook, 2008*; *Stuber et al., 2010*; *Sulzer et al., 1998*; *Tecuapetla et al., 2010*; *Tritsch et al., 2012*). In *Drosophila*, the terminals of the MB DANs contain both clear and dense-core vesicles (*Takemura et al., 2017*), prompting us to ask if the DAN cell types innervating different MB compartments might use distinct cotransmitters that could play a role in generating compartment-specific learning rules. We individually activated several DAN cell types in a tyrosine hydroxylase (TH) mutant background that

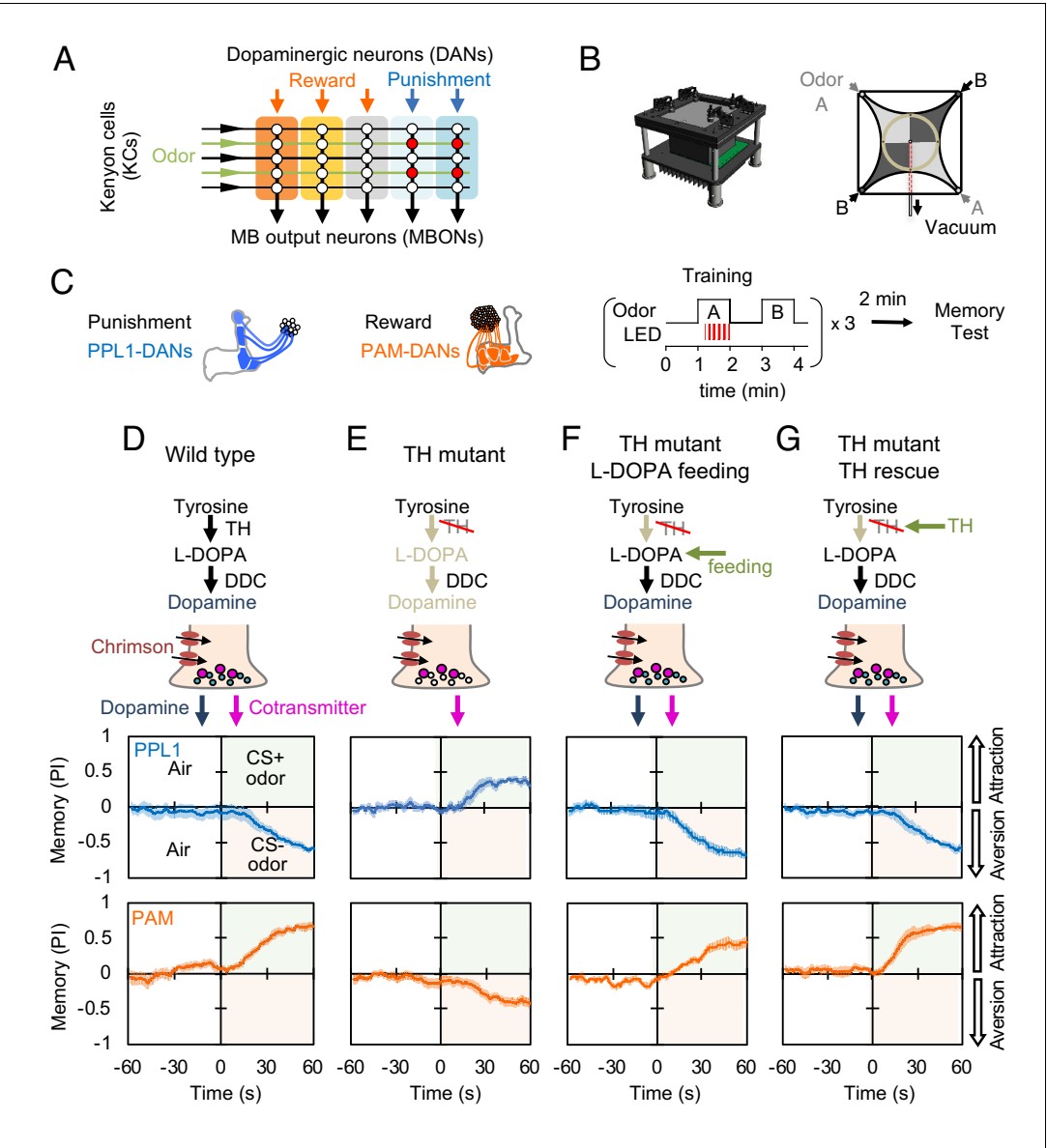

**Figure 1.** Dopaminergic neurons can induce memories without dopamine, but with opposite valence. (A) Conceptual diagram of the circuit organization in the MB lobes. Sparse activity in the parallel axonal fibers of the KCs represent odor stimuli. DANs induce plasticity at KC to MBON synapses (represented by circles), when DAN and KC activity are coincident (red circles). The MB compartments (indicated by the colored rectangles) differ in their learning and memory decay rates. The actual MB lobes contains 15 compartments (*Tanaka et al., 2008*). (B) Design of the optogenetic olfactory arena and a diagram illustrating odor paths in the arena. (C) Schematic representation of the innervation patterns of the PPL1 (blue; *R52H03-p65ADZp; TH-ZpGAL4DBD*) and PAM cluster (orange; *R58E02-p65ADZp; DDC-ZpGAL4DBD*) DANs used to train flies. A diagram of the training protocol is also shown. Flies were trained and tested in the olfactory area. A 1 min odor exposure was paired with thirty 1 s pulses of red light (627 nm peak and 34.9 µW/mm$^2$), followed by 1 min without odor or red light, and then presentation of a second odor for 1 min without red light. In one group of flies, odors A and B were 3-octanol and 4-methylcyclohexanol, respectively, while in a second group of flies, the odors were reversed. Memory was tested immediately after three repetitions of training bouts by giving flies a binary choice between the two odors in the olfactory area. (D–G) Odor memories induced by the collective optogenetic activation of PPL1-γ1pedc, PPL1-γ2α'1, PPL1-α'2α2 and PPL1-α3 DANs (upper panels; blue lines) or activation of PAM cluster DANs (lower panels; orange lines) in wild type (D), TH mutant (E), TH mutant feed 1 mg/ml L-DOPA and 0.1 mg/ml carbidopa (F), or TH mutant with cell-type specific expression of a wild-type TH cDNA (G; see the Materials and methods for drug treatment and supplemental information for genotypes). Time courses of the performance index (PI) during the test period are shown as the average of reciprocal experiments. The PI is defined as [(number of flies in the odor A quadrants) - (number of flies in odor B quadrants)]/ (total number of flies). Thick line and shading represent mean ± SEM. N = 12–16. Two split-GAL4 drivers R52H03-p65ADZp in attP40; TH-ZpGAL4DBD in VK00027 and R58E02-p65ADZp in attP40; DDC-ZpGAL4DBD in VK00027 were used for driving 20xUAS-CsChrimon-mVenus in PPL1 or PAM DANs, respectively.

The online version of this article includes the following figure supplement(s) for figure 1:

*Figure 1 continued on next page*

*Figure 1 continued*

**Figure supplement 1.** Duration of L-Dopa and Carbidopa feeding.
**Figure supplement 2.** Cell-type-specific rescue of TH.

eliminates dopamine synthesis in the nervous system (*Cichewicz et al., 2017*; *Riemensperger et al., 2011*) and assayed their ability to induce associative learning when paired with an odor stimulus.

We first examined the PPL1 cluster of DANs, which innervate several MB compartments involved in aversive learning, driven by stimuli such as electric shock, noxious temperature, and bitter taste (*Galili et al., 2014*; *Kirkhart and Scott, 2015*; *Mao and Davis, 2009*; *Riemensperger et al., 2005*; *Tomchik, 2013*). Using an optogenetic olfactory arena (*Aso and Rubin, 2016*), we trained flies by pairing odor exposure with optogenetic activation of these DANs using CsChrimson-mVenus and then immediately tested memory (*Figure 1B,C*). In flies with a wild-type TH allele, this training protocol induced robust negative-valence memory of the paired odor (*Figure 1D*), as observed previously (*Claridge-Chang et al., 2009*; *Schroll et al., 2006*). In the dopamine-deficient background, activation of the same DANs still induced a robust odor memory, but its valence was now positive (*Figure 1E*). This result is consistent with previous findings that TH mutant flies show a weak positive-valence memory after odor-shock conditioning (*Riemensperger et al., 2011*), although the positive-valence memory we observed is much stronger.

Arguing against the possibility that this valence-inversion phenotype resulted from a developmental defect caused by the constitutive absence of dopamine (*Niens et al., 2017*), feeding L-DOPA and carbidopa to adult-stage flies fully restored normal valence memory (*Figure 1F* and *Figure 1—figure supplement 1*). Nor did the valence-inversion phenotype result from lack of dopamine signaling outside the MB, as restoring TH expression specifically in the PPL1 DANs was sufficient to restore formation of negative-valence memory (*Figure 1G*; *Figure 1—figure supplement 2*). Moreover, valence-inversion in the absence of dopamine was not limited to punishment-representing DANs. Activation of reward-representing PAM cluster DANs (*Burke et al., 2012*; *Liu et al., 2012*) in the TH mutant background also induced odor memory of opposite valence, in this case negative rather than positive (*Figure 1D,E*); as we found for PPL1-induced memories, either L-DOPA plus carbidopa feeding or TH expression in reward-representing DANs restored the ability to form a memory of the valence that is observed in wild-type flies (*Figure 1F,G*). These observations suggested the possible presence of a cotransmitter in these DANs that exerts an opposite effect from dopamine on synaptic plasticity and memory.

## Putative cotransmitter effects differ among DAN cell types

If DAN cell types use different cotransmitters, we might expect the effects of activating DANs in the TH mutant background to vary with cell type. We tested this idea by comparing the associative memories formed in wild-type and TH mutant backgrounds when an odor was paired with optogenetic activation of different subsets of DAN cell types using seven driver lines (*Figure 2* and *Figure 2—figure supplement 1*). We also demonstrated that valence inversion is not limited to training using direct DAN stimulation with CsChrimson; activating bitter taste sensory neurons using Gr66a-GAL4, which activate PPL1-DANs (*Das et al., 2014*; *Kirkhart and Scott, 2015*), likewise induced memories of inverted valence (*Figure 2*). We identified two DAN cell types that exhibited the valence-inversion phenotype: PPL1-γ1pedc and PAM-γ5/PAM-β′2a. With PPL1-γ1pedc stimulation, memory valence switched from negative to positive in TH mutant animals. Conversely, with PAM-γ5 and PAM-β′2a stimulation, memory valence flipped in the opposite direction, from positive to negative in TH mutants.

The valence-inversion phenotype was not, however, observed in all compartments. Activation of PPL1-γ2α′1 resulted in negative-valence memory in both TH mutant and wild type, suggesting a cotransmitter with the same sign of action as dopamine. Activation of PPL1-α3 or PAM-β′1 in the TH mutant background did not induce significant memory, indicating that these cells do not express a cotransmitter capable of inducing memory without dopamine.

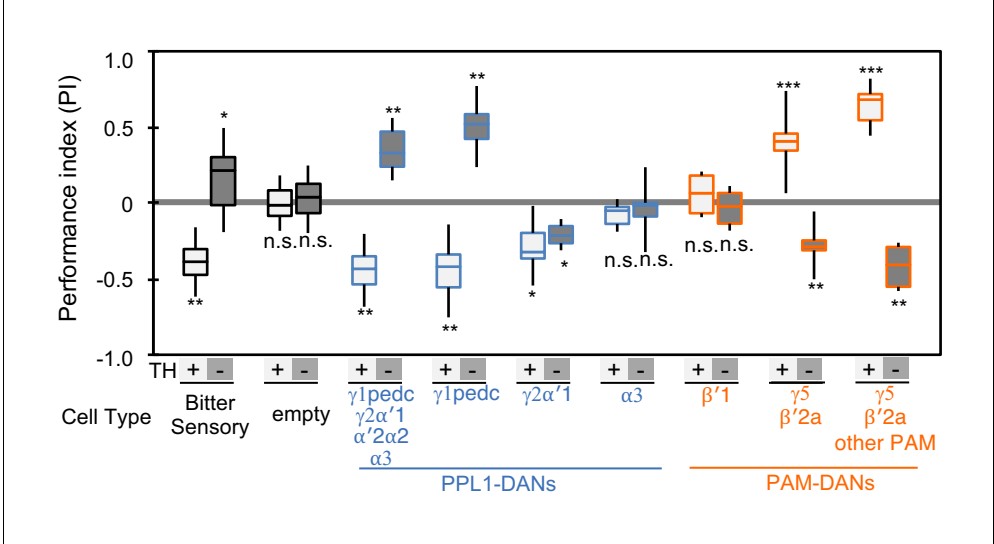

**Figure 2.** Cotransmitter effects differ among DAN cell types. The PI of olfactory memories measured immediately after 3 × 1 min training in TH mutant/TH+ heterozygotes (+) or TH mutant/TH mutant (-) background. Odor presentation was paired with either activation of bitter taste neurons (Gr66a-GAL4) or the indicated subset of DANs using the optogenetic activation protocol diagrammed in *Figure 1C*. A split-GAL4 driver without enhancer (empty) was used as a control. The bottom and top of each box represents the first and third quartile, and the horizontal line dividing the box is the median; the PI was calculated by averaging the PIs from the final 30 s of each test period (see legend to *Figure 1*). The whiskers represent the minimum and maximum. N = 8–16. Asterisk indicates significance from the empty-GAL4 control. Comparison with chance level (i.e. PI = 0) resulted in identical statistical significance: *, p<0.05; **, p<0.01; ***, p<0.001; n.s., not significant. See *Figure 2—figure supplement 1* and www.janelia.org/split-gal4 for expression patterns of split-GAL4 drivers.

The online version of this article includes the following figure supplement(s) for figure 2:

**Figure supplement 1.** Expression patterns of driver lines.

## Identification of nitric oxide synthase in a subset of DAN cell types by transcript profiling

To identify potential cotransmitters, we profiled (using RNA-Seq methods) the transcriptomes of the DAN cell types in these seven split-GAL4 lines, and looked for candidates whose expression correlated with the valence-inversion effect. Isolation of pure populations of specific DAN cell types is challenging because of their low abundance. For example, PPL1-γ1pedc is a single cell in each brain hemisphere and thus requires ~50,000 fold enrichment. We used a collection of split-GAL4 driver lines (*Aso et al., 2014a*) to fluorescently mark the soma of specific DAN cell types and confirmed that the nuclear-targeted reporters we used for sorting visualized the same restricted set of cells as the membrane-targeted reporters in the original study. In this way, we selected a combination of reporter and driver lines that provided the most specific labeling of the targeted cell type. The number of detected genes and the correlation across biological replicates of RNA profiling experiments has been observed to be highly dependent on cDNA yield during library construction (*Davis et al., 2018*). Due to different soma sizes, the amount of mRNA per cell is expected to differ across cell types. To estimate the number of cells necessary for our experimental condition, we started with MBON-γ1pedc>α/β cells, a cell type that occurs once per hemisphere. Three replicates with ~350 cells yielded on average 5.3 μg of cDNA, and we observed a high correlation across biological replicates on these dataset (Pearson R = 0.90). Thus, we aimed for similar cDNA yields by sorting more cells for cell types with a smaller soma (i.e. KCs and PAM cluster DANs). We collected data from 10 driver lines for DAN cell types, with two to four biological replicates per line. We also examined three classes of KCs, DPM, APL and five additional MBON cell types. On average, we collected 1,000–2,500 cells for KCs, 56+ /- 18 for DPM and APL, 546 + /- 60 cells for PAM DANs, and 296 + /- 15 cells for PPL1 DAN and MBON cell types per replicate, by hand or fluorescence activated cell sorting (FACS), yielding 3.39 + /- 0.21 μg of cDNA, 17.7 + /- 1.2 million mapped reads per

replicate, and 0.80 + /- 0.02 Pearson R across biological replicates (*Figure 3—figure supplement 1*; *Supplementary file 1*).

We analyzed these data for different splicing isoforms (see Materials and methods). Using DEseq2 (*Love et al., 2014*), we searched for transcripts that were differentially expressed among DAN cell types and, in particular, for those commonly expressed in DAN cell types that showed the valence-inversion phenotype (PPL1-γ1pedc and PAM-γ5), but not in other cell types (*Figure 3A,B*). Through this analysis, we identified nitric oxide synthase (NOS) as a strong candidate for an enzyme synthesizing a cotransmitter. Similar enrichment in PPL1-γ1pedc and PAM-γ5 was found in only five other transcripts, none of which are likely to encode a neurotransmitter: (i) *epac-RG*, a cAMP-activated guanine nucleotide exchange factor. (ii) *br-RO, br-RI,* both transcripts of the zinc finger transcription factor *broad*, (iii) *CG32547-RD*, a G-protein-coupled receptor, and (iv) *CG12717-RA*, a SUMO-specific isopeptidase (data reviewed in FlyBase) (*Thurmond et al., 2019*). In addition to these transcripts that matched our criteria, we found other potential candidates whose expression was not a precise match. We detected a high level of the *DH44* neuropeptide in PPL1-γ1pedc, but not in PAM-γ5. A receptor for DH44, *DH44-R1*, was expressed in PAM-γ4 and/or γ4<γ1γ2 and to lower extent in α/β Kenyon cells, but neither of the two known receptors for DH44 was detected in γ Kenyon cells (*Figure 3—figure supplement 9*). Transcripts of the neuropeptide gene *Nplp1* were detected in PPL1-γ1pedc and PAM-γ5, but other DANs and MBONs also expressed this gene (*Figure 3—figure supplement 8*) (*Croset et al., 2018*). Expression of *Gyc76C*, the receptor for Nplp1, was barely detectable in KCs, DANs, and MBONs. Complete transcript data were deposited to NCBI Gene Expression Omnibus (accession number GSE139889) and presented in *Supplementary file 1*, and the expression of genes encoding neurotransmitters, neuropeptides and their receptors, as well as components of gap junctions, is summarized in *Figure 3—figure supplement 2–10*. Although we cannot formally rule out a contribution of other genes and pathways, we chose to pursue NOS, as it was the most promising candidate gene for producing a cotransmitter that might be responsible for the valence-inversion effect.

*Drosophila* has only one gene encoding nitric oxide synthase (*Nos*), but this gene has multiple splicing isoforms (*Figure 3C*) (*Rabinovich et al., 2016*; *Regulski and Tully, 1995*; *Stasiv et al., 2001*). Only NOS1, the full-length isoform, is functional, while the truncated isoforms can function as a dominant-negative. Thus, identifying the expressed splicing isoform of *Nos* was crucial for understanding NOS functions in DANs. NOS1 was the most abundantly expressed isoform as judged by RNA profiling. We confirmed NOS1 expression by combining fluorescent in situ hybridization (FISH) and antibody staining. For whole-brain FISH (*Long et al., 2017*), we used 40 probes against c-terminus exons that are present in NOS1 and NOS4, but not NOS-RK, transcripts (*Figure 3C* and *Figure 3—figure supplement 1D*). PPL1-γ1pedc was labeled with these FISH probes (*Figure 3—figure supplement 1D*), confirming expression of NOS1 or NOS4 in these cells. For immunohistochemistry, we used an antibody raised against exon 16 of NOS (*Kuntz et al., 2017*; *Yakubovich et al., 2010*) that is present in NOS1 and NOS-RK, but not in NOS4 (*Figure 3C*). We validated its specificity by demonstrating a loss of the staining that accompanied RNAi-mediated knockdown of NOS (*Ni et al., 2011*) in PPL1-γ1pedc (*Figure 3D* and *Figure 3—figure supplement 1B*). In the MB lobes, the γ1pedc and γ5 compartments showed enriched anti-NOS immunoreactivity, as expected from the RNA-Seq data (*Figure 3E*). Furthermore, we confirmed colocalization of NOS with γ1pedc's terminals by expansion microscopy and lattice light sheet microscopy (ExM LLSM; *Figure 3F and G*, *Video 1*) (*Chen et al., 2015*; *Gao et al., 2019*; *Tillberg et al., 2016*). Although the majority of anti-NOS immunoreactivity in γ1 compartment was diminished by expressing NOS-shRNA in PPL1-γ1pedc, we noticed sparse but large NOS-positive puncta remained (*Figure 3D*). Similar large structures were observed outside the DAN's terminals with ExM LSSM and were found to colocalize with terminals of octopaminergic neurons (*Figure 3F* right). In addition, γ3 and γ4 also showed significant anti-NOS immunoreactivity (*Figure 3E*). However, the low NOS transcript levels observed via RNA-Seq in PAM-γ3 and PAM-γ4 are most consistent with anti-NOS immunoreactivity in γ3 and γ4 arising from non-DAN cell types or developmental expression. The cell-type-specific expression and localization of NOS1 in a subset of DANs associated with compartments that display the valence-inversion phenotype prompted us to test the hypothesis that NO plays a direct role in generating the diversity of memory dynamics observed in different compartments.

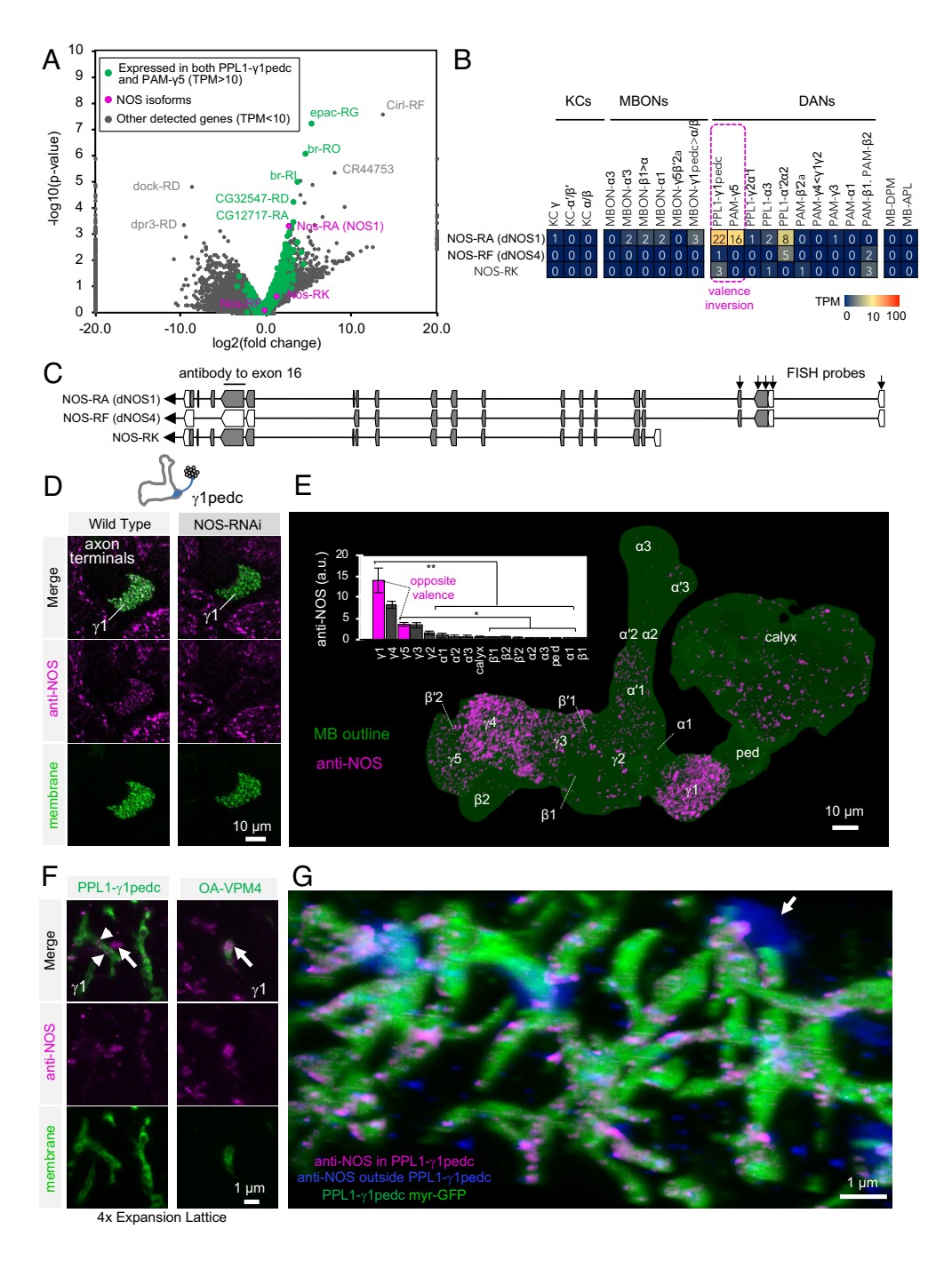

**Figure 3.** Identification of NOS1 in PPL1-γ1pedc and PAM-γ5 by RNA-Seq. (**A**) The RNA-Seq data of two cell types that showed the valence-inversion phenotype (i.e. PPL1-γ1pedc and PAM-γ5) were pooled and compared against the pooled data of all other cell types examined (DANs, MBONs, KCs). The -$\log_{10}$ of p-values for comparing number of counted transcripts between the pooled data of PPL1-γ1pedc and PAM-γ5 relative to that of all other cell types were plotted against the $\log_2$ of fold changes observed in the expression levels of transcripts. Different splice isoforms of genes are plotted separately (dots). Green dots represent the 2981 transcripts expressed at levels above 10 transcripts per million (TPM) in both PPL1-γ1pedc and PAM-γ5. Gray dots represent the 31,539 transcripts with expression levels below 10 TPM in one or both of these two DANs. Dots on +20 or −20 x-axis represent splice isoforms that were detected only in the pooled PPL1-γ1pedc and PAM-γ5 or in other cell types. Magenta dots show the three splice isoforms of NOS. (**B**) Mean TPM of NOS splicing isoforms. The magenta dashed line highlights PPL1-γ1pedc and PAM-γ5, the two cell types that showed the valence-inversion phenotype. (**C**) Map of the NOS locus. Exons and protein coding sequences are depicted as boxes and gray boxes, respectively. Only the full-length isoform NOS-RA (dNOS1) produces a functional NOS protein (*Stasiv et al., 2001*). The antibody we used was raised

*Figure 3 continued on next page*

*Figure 3 continued*

against exon16 (*Kuntz et al., 2017*; *Yakubovich et al., 2010*). Arrows indicate the first four exons of NOS1 and NOS4 where 40 FISH probes were designed to recognize these, but not NOS-RK transcripts. See Materials and methods for details of the position and sequence of the probes. (**D**) NOS immunoreactivity was observed in the γ1pedc compartment of the MB. Immunoreactivity was markedly reduced by expressing NOS-RNAi in PPL1-γ1pedc. (**E**) Distribution of NOS-immunoreactivity inside the MB is displayed. Voxels above mean +2SD of the entire brain are shown in 12-bit scale in magenta. The insert shows a quantification of NOS-immunoreactivity in each MB compartment. Signal in γ1 was significantly higher than 12 compartments indicated by the bracket (Kruskal-Wallis with Dann's test for selected pairs). Signal in γ5 was significantly higher than eight compartments indicated by the bracket; *, p<0.05; **, p<0.01; n = 12. (**F**) Lattice light sheet image of a 4x expanded brain shows colocalization of NOS immunoreactivity and the terminals of PPL1-γ1pedc (left; arrowheads). In addition, large but sparse NOS puncta were observed outside PPL1-γ1pedc (left; arrow); these puncta match the pattern of terminals of OA-VPM3 and/or OA-VPM4 (right; arrow), indicating that these octopaminergic neurons also expresses NOS. (**G**) 3D reconstruction of lattice light sheet image of the γ1 with pseudo colors for NOS inside (magenta) or outside (blue) the PPL1-γ1pedc DANs (green). The arrow indicates a large NOS puncta outside PPL1-γ1pedc.

The online version of this article includes the following figure supplement(s) for figure 3:

**Figure supplement 1.** Controls for RNA-Seq reproducibility, anti-NOS antibody specificity and FISH probes.
**Figure supplement 2.** RNA-seq data for genes related to neurotransmitter synthesis and transporters.
**Figure supplement 3.** RNA-seq data for acetylcholine and GABA receptors.
**Figure supplement 4.** RNA-seq data for glutamate, glycine and histidine receptors.
**Figure supplement 5.** RNA-seq data for monoamine receptors.
**Figure supplement 6.** RNA-seq data for gap junctions and neuropeptide processing.
**Figure supplement 7.** RNA-seq data for neuropeptides.
**Figure supplement 8.** RNA-seq data for neuropeptides (continuation of supplement 7).
**Figure supplement 9.** RNA-seq data for neuropeptide receptors.
**Figure supplement 10.** RNA-seq data for neuropeptide receptors.

## NOS in dopaminergic neurons contributes to memory formation

We next evaluated the role of NOS in memory formation in the absence of dopamine biosynthesis. If NO is indeed the cotransmitter that supports the valence-inverted memory in TH mutant flies, we would predict that inhibiting NOS should block this effect, and that flies would show no memory. To assess the requirement for NO synthesis, we fed flies the competitive NOS inhibitor L-NNA for one day before training and then measured the memory induced by optogenetic training using PPL1-γ1pedc in a TH mutant background. We found that this treatment reduced valence-inverted memory formation in a dose-dependent manner (*Figure 4A*). Whereas NOS is broadly expressed in the brain (*Kuntz et al., 2017*), two lines of evidence suggest that L-NNA fed flies are capable of olfactory learning. First, when we bypassed the TH mutant by feeding L-DOPA and carbidopa to restore dopamine levels, the L-NNA fed flies showed a normal level of negative-valence odor memory formation in response to pairing an odor with PPL1- γ1pedc activation (*Figure 4A*). Second, the effect of L-NNA feeding was cell-type specific; memory formed by activation of either PPL1-γ1pedc or PAM-γ5/β′2a was affected but that formed by activation of PPL1-γ2α′1, was not (*Figure 4B*). We obtained consistent results in knockdown experiments where we expressed NOS-RNAi in all PPL1 DANs (*Figure 4C*). In this case, the negative-valence memory observed with NOS-RNAi in the TH mutant background is consistent with the presence of a different cotransmitter in PPL1-γ2α′1 (*Figure 2*).

We also examined whether we could transfer this valence-inversion property to another compartment by ectopically expressing NOS. We expressed NOS in PPL1-α3, a compartment where we observed no intrinsic NOS expression, and where extended optogenetic training induces a negative-valence memory (*Aso and Rubin, 2016*). TH mutant flies formed no odor association with PPL1-α3 activation, but when NOS was ectopically expressed in the α3 compartment,

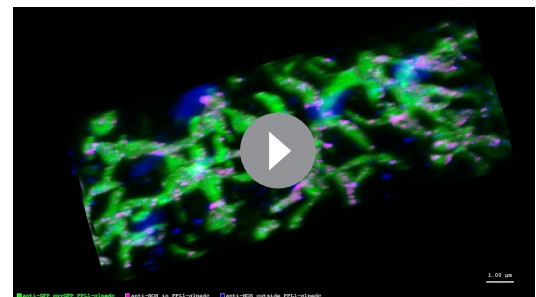

**Video 1.** Anti-NOS signals inside or outside the PPL1-γ1pedc. Rotation movie of the 3D reconstruction of lattice light sheet image of DANs in γ1 (green) with pseudo colors for NOS inside (magenta) or outside (blue).
https://elifesciences.org/articles/49257#video1

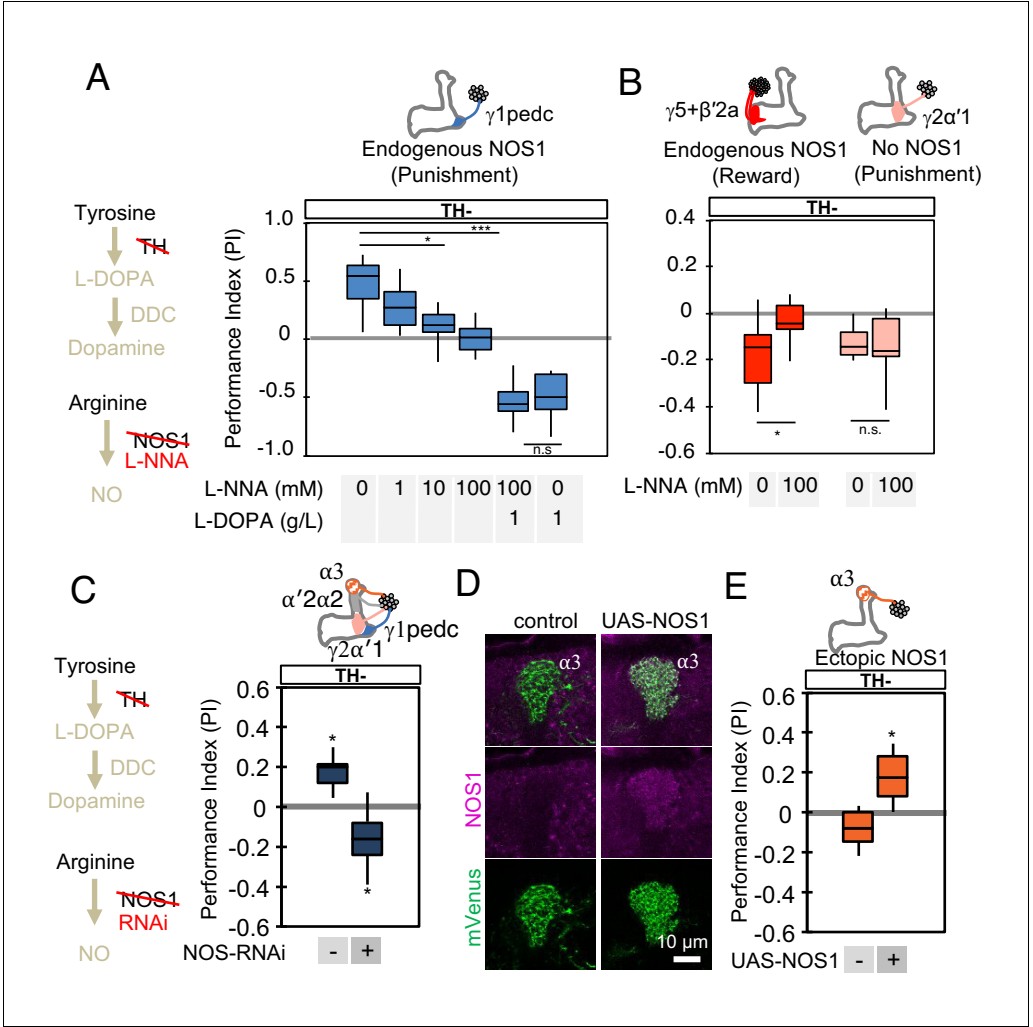

Figure 4. NOS in DANs contributes to memory formation. (A) Increasing the dose of L-NNA reduced the positive-valence memory induced by activation of PPL1-γ1pedc in a TH mutant background. The ability to form an negative-valence memory was restored by feeding of L-DOPA plus carbidopa and this memory formation was not affected by L-NNA. N = 8–12. (B) Feeding of L-NNA in a TH mutant background reduced the negative-valence memory induced by activation of the combination of PAM- γ5 and PAM-β′2a, but not of PPL1-γ2α′1. N = 12–16 (C) Activation of PPL1 DANs (PPL1-γ1pedc, PPL1-γ2α′1, PPL1-α′2α2 and PPL1-α3) induced significant positive-valence memory in a TH mutant background. The valence of the induced memory was negative when NOS-RNAi was expressed in the same DANs. We postulate that the negative-valence memory observed when NOS-RNAi is expressed results from an as yet unidentified cotransmitter released by PPL1-γ2α′1 (see also panel B and Figure 2). N = 8 (D) NOS immunoreactivity in the α3 compartment in wild type (left) and after ectopic expression of NOS (right). (E) Activation of PPL1-α3 in which NOS was ectopically expressed induced significant positive-valence memory after 3 × 1 min training protocol in a TH mutant background (Figure 1C). Note that activation of PPL1-α3 can induce negative-valence memory in a wild-type background, but only after 10x spaced training (Aso and Rubin, 2016). N = 12 In A-C and E, memories assessed immediately after 3 × 1 min training are shown. The bottom and top of each box represents the first and third quartile, and the horizontal line dividing the box is the median. The whiskers represent the minimum and maximum. N = 8–16. Asterisk indicates significance of designated pair in A and B, or from 0 in C and D: *, p<0.05; ***, p<0.001; n.s., not significant.

The online version of this article includes the following figure supplement(s) for figure 4:

Figure supplement 1. Ectopic expression of NOS in PPL1-γ2α′1 can change the valence of the memory formed by this cell type.

the same training protocol induced a positive-valence memory (*Figure 4D and E*). Similarly, ectopic expression of NOS in PPL1-γ2α′1 also resulted in valence inversion phenotype (*Figure 4—figure supplement 1*). In other words, NO was able to form an association of opposite valence to that formed by dopamine. These results demonstrate the functional significance of NOS in DANs, but leave open its mechanism of action.

## Soluble guanylate cyclase in Kenyon cells is required to form NO-dependent memories

In the MB, dopamine induces synaptic plasticity by binding to dopamine receptors on the axons of KCs and activating the *rutabaga*-encoded adenylyl cyclase in these cells (*Gervasi et al., 2010*; *Tomchik and Davis, 2009*). Does NO released from DANs also act on receptors in the KCs? RNA-seq data revealed expression of the subunits of soluble guanylate cyclase (sGC), Gycα99B and Gycβ100B, in KCs, DANs, and MBONs (*Figure 5A–B*). Similar to its mammalian homologs, *Drosophila* sGC is activated upon binding NO (*Morton et al., 2005*). Transcripts for other guanylyl cyclases were found at lower levels, if at all (*Figure 5A–B*; *Supplementary file 1*), suggesting that sGC formed by Gycα99B and Gycβ100B is the primary source of cGMP in KCs. Consistent with these transcriptomic data, we observed that a protein trap Gycβ100B-EGFP fusion protein (MI08892; *Venken et al., 2011*) was broadly expressed in the MB lobes (*Figure 5—figure supplement 1*).

We tested the role of Gycβ100B with acute RNAi knockdown in KCs using the MB-switch system (*McGuire et al., 2003*). This RNAi knockdown effectively reduced expression of endogenous Gycβ100B tagged with GFP in the MB (*Figure 5—figure supplement 1C*). We found that a reduction in Gycβ100B produced in this way abolished the valence-inverted memory induced by activation of PPL1-γ1pedc in a TH mutant background (*Figure 5C* and *Figure 5—figure supplement 1D*). Although MB-switch showed significant leaky expression without RU-486 feeding under our experimental conditions (*Figure 5—figure supplement 1E and F*), we could restore normal aversive memory by feeding TH mutant flies L-DOPA (*Figure 5C*), indicating that flies carrying MB-switch driven Gycβ100B-RNAi do not have a general defect in learning. Taken together, our data argue strongly that NO functions as cotransmitter that is released by DANs and acts on sGC in postsynaptic KCs to regulate cGMP levels, although our results do not exclude the possibility that NO has other targets. Indeed, both MBONs and DANs express sGC (*Figure 5B*) and we found that optogenetic stimulation of PPL1-γ1pedc in a TH-mutant background can induce a prolonged increase in calcium levels in MBON-γ1pedc>α/β, which was suppressed by L-NNA (*Figure 5—figure supplement 2A*). However, DANs expressing sGC in other compartments did not respond to stimulation of PPL1-γ1pedc in a TH-mutant background (*Figure 5—figure supplement 2B*), raising the possibility that NO may not be able to diffuse over the 30 µm distances required to reach those compartments.

## NO-dependent and dopamine-dependent memories have different kinetics

The above results establish the functional significance of NOS in a TH mutant background, but leave open the question of what role NO signaling plays in memory formation in wild-type flies. PPL1-γ 1pedc induces a valence inverted memory when its activity precedes an odor such that the odor predicts relief from punishment (*Aso and Rubin, 2016*; *Tanimoto et al., 2004*). NOS is dispensable for this timing dependent inversion of memory valence (*Figure 5—figure supplement 3*), an observation consistent with a recent report that timing dependent inversion of long-term-depression of KC-to-MBON synapses to long-term-potentiation is mediated by different dopamine receptors (*Handler et al., 2019*). Next, we examined if NO-dependent memory requires, *scribble* (*scrib*), a gene encoding a scaffold protein which is important for forgetting; loss of scribble in KCs prolongs memory retention of odor-electric shock associative memories (*Cervantes-Sandoval et al., 2016*). RNAi knockdown of *scrib* impaired NO-dependent memory in TH-mutant flies (*Figure 5D*), suggesting a role of NO in regulating memory stability.

To further explore the interplay of dopamine and NO signaling on memory dynamics we examined the consequences of their combined action. First, we examined memory acquisition rates when flies were trained by activation of PPL1-γ1pedc (*Figure 6A*). In wild-type flies, PPL1-γ1pedc activation as brief as 10 s can induce significant negative-valence memory. Blocking NOS activity with L-NNA did not affect the memory scores observed shortly after a wide range of training protocols

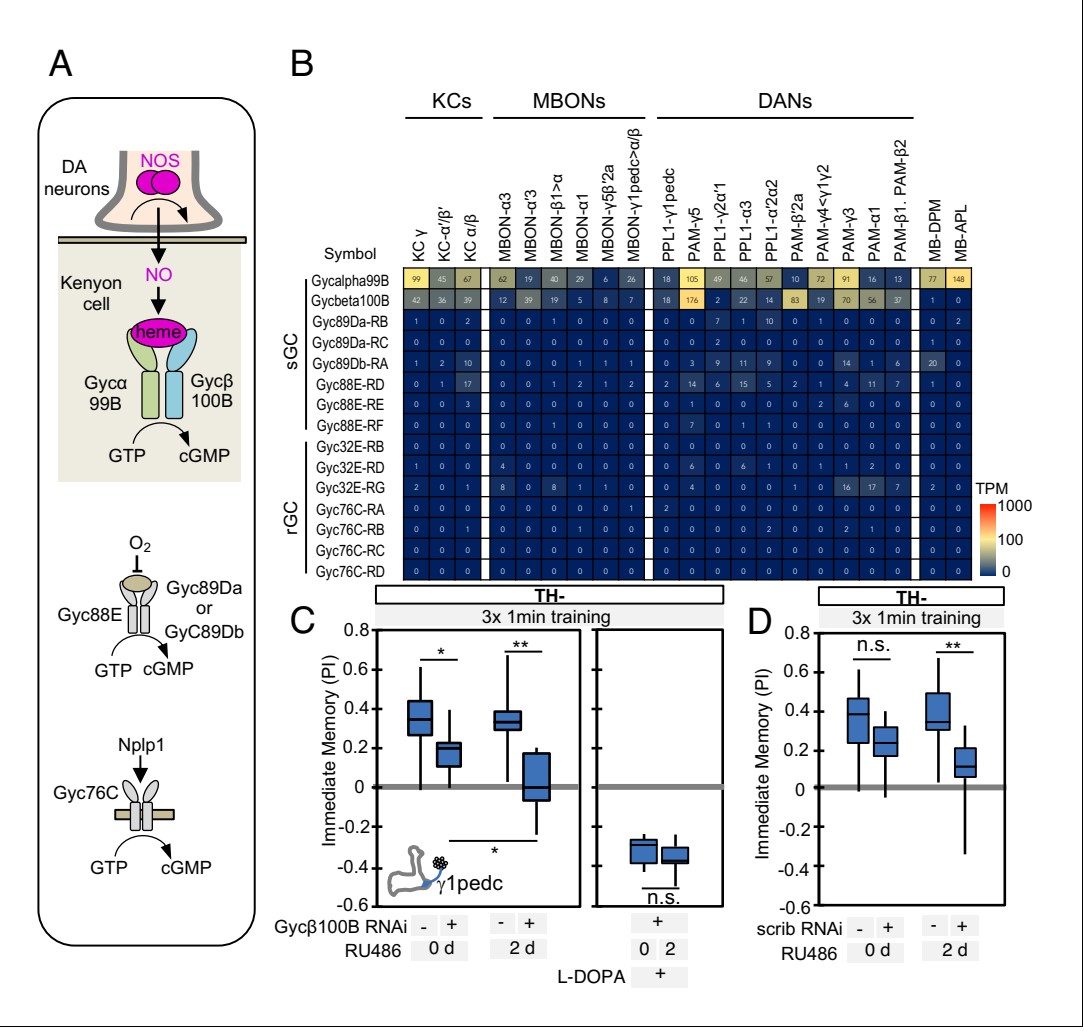

**Figure 5.** Soluble guanylate cyclase in the KCs is required to form NO-dependent memory. (**A**) Diagram of soluble or receptor guanylyl cyclases in *Drosophila*. (**B**) RNA-seq data indicate coexpression of Gycα99B and Gycβ100B in KCs, MBONs and DANs. For comparison, the expression levels of other guanylate cyclase genes are also shown. Note that RNA-Seq detected transcripts of neuropeptide gene Nplp1 in both PPL1-γ1pedc and PAM-γ5 (***Figure 3—figure supplement 8***), but expression of its receptor Gyc76C was barely detectable compared to Gycα99B and Gycβ100B. (**C**) Induction of Gycβ100B-shRNA in Kenyon cells by activating MB247-switch driver (***Mao et al., 2004***) with RU-486 feeding reduced the positive-valence memory induced by PPL1-γ1pedc. We also observed a partial effect in the flies without RU-486, presumably due to leaky expression (***Figure 5—figure supplement 1E and F***). Negative-valence memory with additional feeding of L-DOPA and carbidopa was not affected by Gycβ100B-shRNA induction in KCs. Memories immediately after 3 × 1 min training are shown. The bottom and top of each box represents the first and third quartile, and the horizontal line dividing the box is the median. (**D**) Induction of scrib-shRNA in KCs also reduced the positive-valence memory induced by activation of PPL1-γ1pedc in a TH mutant background. The whiskers represent the minimum and maximum. N = 12–16. Asterisk indicates significance of designated pair: *, p<0.05; **,p<0.01; n.s., not significant.

The online version of this article includes the following figure supplement(s) for figure 5:

**Figure supplement 1.** Expression of Gycbeta100B in the mushroom body lobes (**A**) Distribution of Gycbeta100B-EGFP in flies carrying the Gycbeta100B[MI08892-GFSTF.2] construct in the MB lobes is shown in a series of anterior to posterior confocal sections.

**Figure supplement 2.** NO from PPL1-γ1pedc activates MBON-γ1pedc but not PAM-DANs in γ3, γ4 and γ5.

**Figure supplement 3.** NO is not involved in timing-dependent inversion of valence.

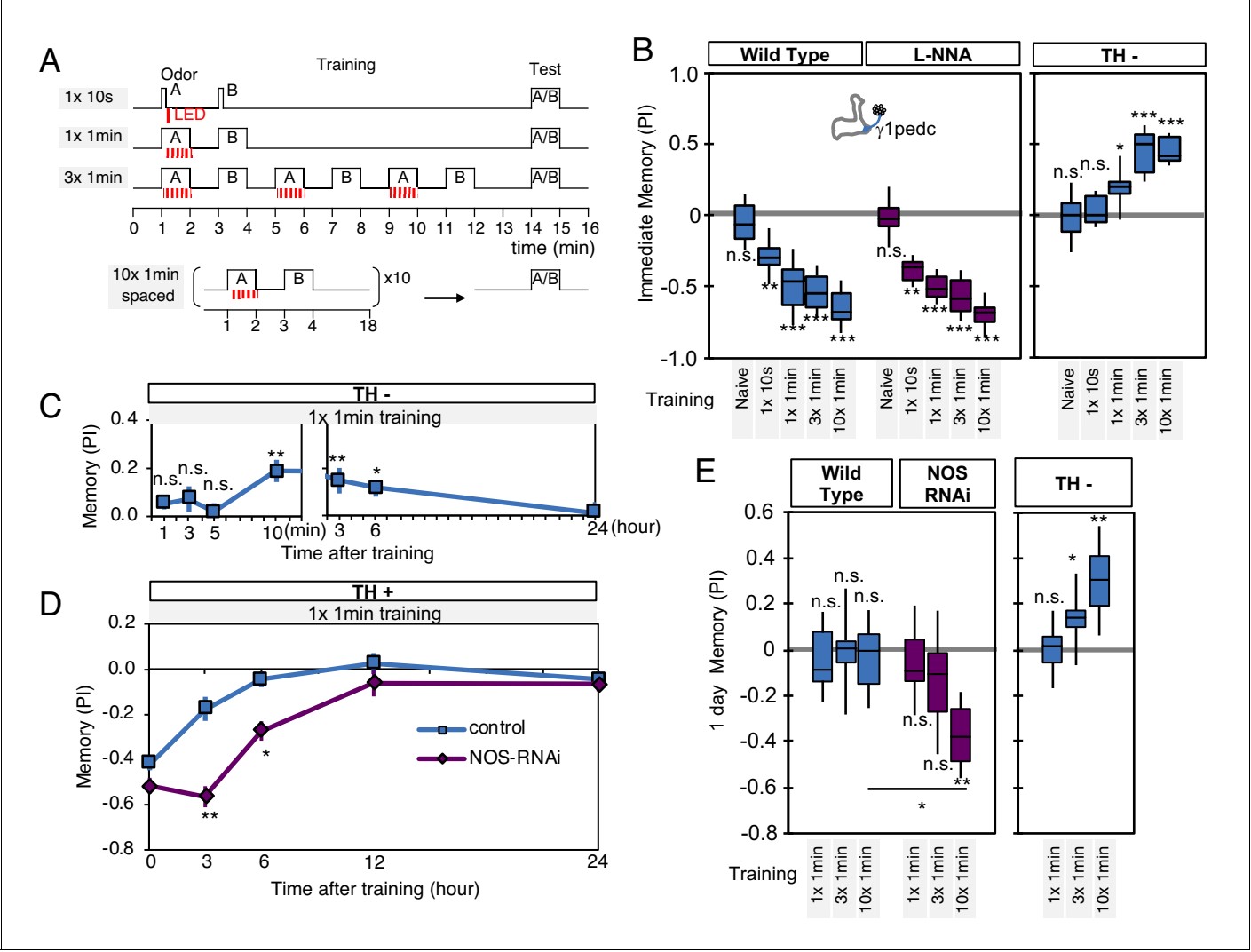

**Figure 6.** NO-dependent effect develops slowly, requires longer training than dopamine-dependent memory, and shortens memory retention. (**A**) Schematic diagram of training protocols. (**B**) Learning rate by activation of PPL1-γ1pedc in wild type (left; blue), wild type with L-NNA feeding (center; purple) or TH mutant backgrounds (right; blue). Memory scores were not significantly affected by L-NNA feeding in any of the training protocols in wild-type flies. A single 10 s training was insufficient to induce any detectable memory in a TH mutant background, but induced significant negative-valence memory in wild-type background. (**C**) Time course of NO-dependent, positive-valence memory induced by PPL1-γ1pedc in a TH mutant background after 1 × 1 min training. Data point and error bars indicate mean and SEM. N = 8–10. Note that the plot is split in the time axis to better display the kinetics. (**D**) After 1 × 1 min training, a cell-type-specific knock down of NOS in PPL1-γ1pedc prolonged the retention of negative-valence memory induced by PPL1-γ1pedc in wild-type background measured at 3 and 6 hr. Note that expression of NOS-RNAi did not affect the score of immediate memory. N = 12. (**E**) Effect of repetitive trainings on 1-day memory. Repetitive training with activation of PPL1-γ1pedc did not induced significant 1-day memory in wild-type background irrespective of training protocols (blue; left). Flies expressing NOS-shRNA showed significant 1 day memory after 10x spaced training (purple; center). In a TH mutant background (right), appetitive memory was induced by 3X and 10X repetitive training. For activation of PPL1-γ1pedc, VT045661-LexA was used as the driver for experiments in B, C and in the TH mutant background data in E, and MB320C split-GAL4 was used for wild-type and NOS-RNAi data in D. We made consistent observations with both global L-NNA inhibition of NOS and cell-type-specific NOS-RNAi (see *Figure 6—figure supplement 1*). The bottom and top of each box represents the first and third quartile, and the horizontal line dividing the box is the median. The whiskers represent the minimum and maximum. N = 12–16. Asterisk indicates significance between control and NOS-RNAi in D, between designated pair in E, or from 0 in all others: *, p<0.05; **, p<0.01; ***, p<0.001; n.s., not significant.

The online version of this article includes the following figure supplement(s) for figure 6:

**Figure supplement 1.** Comparison of the effects of NOS-RNAi and L-NNA.

(*Figure 6B*). In contrast, longer and repetitive training was required to induce robust NO-dependent positive-valence memory in the absence of dopamine (*Figure 6A–B*).

*Riemensperger et al. (2011)* reported that dopamine-deficient flies developed weak positive-valence memory after odor-shock conditioning, but this memory was not detectable until 2 hr after the training. Motivated by this observation, we examined the kinetics of NO-dependent memory formation and the role of NOS in memory retention. When we used a single cycle of training, we found that NO-dependent memory develops slowly over time. Memory scores were not significantly different from zero at 1, 3, 5 min after training, and only became significant after 10 min. Once formed, however, these NO-dependent memories were long lasting and were still more than half maximal after 6 hr (*Figure 6C*). This result contrasts with the time course of memory formation by PPL1-γ1pedc activation in a wild-type background, where memory is detectable within 1 min after training but has a half-life of only 2–3 hr (*Aso et al., 2012*). These observations raised the possibility that NO signaling, with its opposite valence and slower dynamics, might serve to limit memory retention in a wild-type background. Indeed, we found that expression of NOS-RNAi or L-NNA feeding prolonged the retention of memories induced by either optogenetic training with PPL1-γ1pedc or odor-shock conditioning (*Figure 6D*, *Figure 6—figure supplement 1A–B*).

Memory persistence is often enhanced by repetitive training. However, PPL1-γ1pedc fails to induce long-lasting memory even after 10x repetitive training at spaced intervals (*Aso and Rubin, 2016*). In contrast, other DANs from the PPL1 cluster that do not exhibit significant NOS expression, PPL1-α3 or a combination of PPL1-γ2α'1 and PPL1-α'2α2, are able to induce stable memory lasting for 4 days after 10x spaced training (*Aso and Rubin, 2016*). Our results strongly suggest that NO signaling is responsible for this difference in memory retention. Spaced training with PPL1-γ1pedc induced memory lasting 1 day when NOS signaling was compromised, either by knockdown with RNAi (*Figure 6E*) or inhibition by L-NNA (*Figure 6—figure supplement 1C*). The valence-inverted memories formed in the γ1pedc compartment following repetitive training in TH mutants also lasted 1 day after training (*Figure 6E*). Thus, the effects of NOS signaling accumulate slowly, but can be long lasting. These effects are antagonistic to memories formed by dopamine signaling, and serve to sculpt the time course of memory retention. As discussed below, NO-signaling also contributes to other features of memory dynamics.

## Nitric oxide enhances fast updating of memory

We designed behavioral experiments to examine memory dynamics when flies that had been previously trained encounter a new experience. We tested three different types of new experience: (1) switching which odor is paired with DAN activation during odor conditioning (reversal conditioning); (2) exposing flies to odors without DAN activation; and (3) activating DANs without odor exposure (*Figure 7A*). In wild-type flies, odor preference can be altered by a single trial of reversal conditioning with PPL1-γ1pedc activation (*Figure 7B*, left), whereas this process became slower and required more training when NOS was inhibited (*Figure 7B*; center), such that switching odor preference required a repetition of reversal conditioning. In TH mutants, NO-dependent memory was also altered by reversal conditioning, but with an even slower time course (*Figure 7B*; right). The second type of new experience, exposure to odors alone, did not change the existing memory in this assay (*Figure 7C*). The third type of new experience, DAN activation alone, quickly reduced conditioned odor response in wild-type flies. Inhibiting NOS slowed this process (*Figure 7D*; center). NO-dependent memory was also reduced by unpaired activation of DANs, but it took five trials to detect significant reduction (*Figure 7D*; right). These results suggest that both the slow formation and the persistence of NO-dependent memory facilitate the fast updating of memories stored in NOS-positive MB compartments in response to changing conditions.

## Modeling the function of NO and dopamine in memory dynamics

To understand the interplay between dopamine (DA) and NO-dependent plasticity, we fit a minimal model to our data that accounts for the observed effects of these two pathways on the formation of odor memories. We then used this model both to infer the parameters of a synaptic plasticity rule consistent with the data as well as to test hypotheses about the mechanisms of DA and NO-mediated synaptic modifications that would be able to generate the memory dynamics we observed. Imaging and physiology experiments have demonstrated that DANs induce intracellular signaling

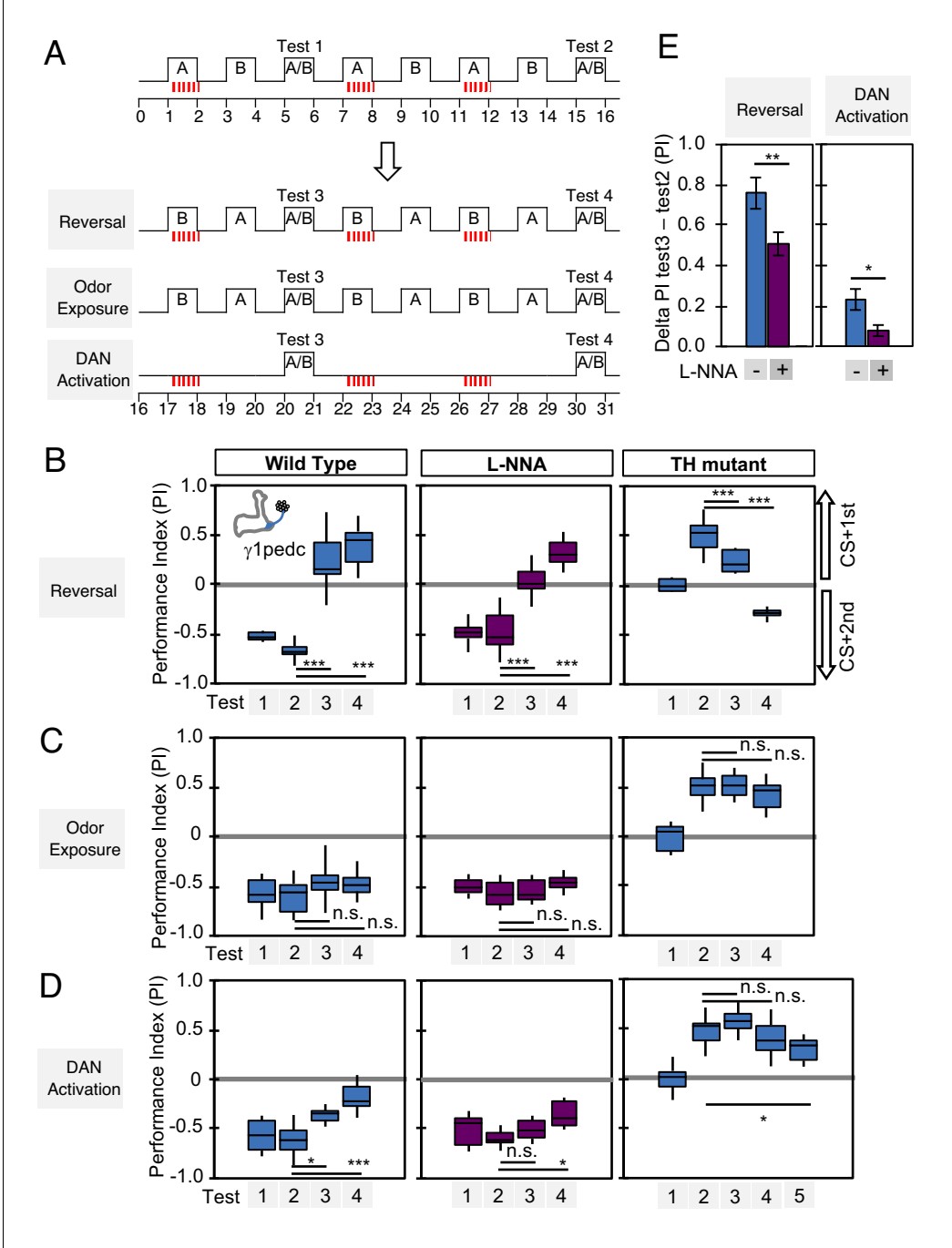

**Figure 7.** Nitric oxide enhances fast update of memory. (**A**) Experimental design to measure dynamics of memory when flies encounter new experiences after establishing an initial odor memory by the 3 × 1 min spaced training pairing odor presentation and optogenetic PPL1-γ1pedc activation. (**B**) In reversal learning, the control odor in the first three trainings was now paired with activation of PPL1-γ1pedc. In all three cases, the first reversal learning was sufficient to modify the odor preference. However, only in wild-type flies was this change large enough that the flies preferred the new odor. In L-NNA fed or TH mutant (dopamine deficient) flies changing the odor preference required multiple training sessions with the new odor. (**C**) Three exposures to each of the two odors did not significantly change the odor preference in any of the three sets of flies. (**D**) One activation of PPL1-γ1ped without odor quickly reduced the conditioned response in wild type. L-NNA fed flies or dopamine deficient flies required three or five times, respectively, repetitions of PPL1-γ1ped activation to significantly reduce the conditioned response. The bottom and top of each box in (**B–D**) represents the first and third quartile, and the horizontal line dividing the box is the median. The whiskers represent the minimum and maximum. (**E**) Changes in PI induced by

*Figure 7 continued on next page*

*Figure 7 continued*

3x training (as measured in Test 2) resulting from the first reversal training (left) or from DAN activation without odor presentation (right) (as measured in Test 3). The observed changes were significantly larger in wild-type flies compared to L-NNA fed flies. *, p<0.05; **, p<0.01.

cascades in KCs and bidirectionally modulate the weights of KC-MBON synapses, with the direction of the plasticity determined by whether each KC is active or inactive (*Cohn et al., 2015*; *Gervasi et al., 2010*; *Hattori et al., 2017*; *Hige et al., 2015*; *Berry et al., 2018*; *Boto et al., 2014*; *Tomchik and Davis, 2009*; *Owald et al., 2015*; *Bouzaiane et al., 2015*; *Cassenaer and Laurent, 2012*; *Handler et al., 2019*; *Okada et al., 2007*). This dependence on presynaptic KC firing ensures the odor-specificity of memories formed following DAN activation. NO-dependent memories, like DA-dependent memories, are odor-specific, suggesting that plasticity induced by NO also depends on KC activity. Based on this observation and the results of previous studies, we constructed our model by assuming that: (1) both DA and NO bidirectionally modulate KC-MBON synapses depending on KC activity, (2) memory decay is due to background levels of DAN activity following conditioning (*Plaçais et al., 2012*; *Berry et al., 2018*; *Sitaraman et al., 2015*), and (3) the effects of DA and NO occur via independent pathways and can coexist (*Figure 6*).

Specifically, we denoted the effects of DA and NO-dependent synaptic plasticity at time $t$ by two quantities, $D(t)$ and $N(t)$, that lie between 0 and 1. We assumed that coincident KC-DAN activation increases $D$ and $N$ with a timescale of 30 s and 10 min, respectively, to account for the slower induction of NO-mediated effects (*Figure 8A,B*). Based on previous observations in TH wild-type flies that pairing of activation of the PPL1-γ1pedc DAN with odor induces synaptic depression between odor activated KCs and MBON-γ1pedc>α/β (*Hige et al., 2015*), we assumed that the effect of an increase in $D$ is a reduction in the weight of the corresponding KC-MBON synapse (*Figure 8B*, left). As the activity of this MBON promotes approach behavior (*Aso et al., 2014b*; *Owald et al., 2015*), its reduced response to the conditioned odor after DA-dependent synaptic depression results in avoidance during subsequent odor presentations. In the TH-null background, in contrast, we have shown that PPL1-γ1pedc activation leads to a positive-valence memory (*Figure 2*). This is most readily explained in our model by assuming that, in such flies, NO induces potentiation of synapses between odor-activated KCs and MBONs (*Figure 8B*, right). Thus, in the model, the effect of an increase in $N$ is an increase in synaptic weight, opposite to the effect of $D$. Finally, based on observations that activation of DANs alone can reverse synaptic depression induced by KC-DAN pairing (*Cohn et al., 2015*; *Hattori et al., 2017*; *Berry et al., 2018*), we assumed that DAN activation in the absence of KC activity causes a reduction in $D$ and $N$, recovering the synapse to its baseline weight.

We fit the model by assuming that the performance index (PI) is determined by the odor-evoked activation of the MBON and then determining the parameters that best match the behavioral data reported in *Figure 6B–D*. We used data that isolates the effects of DA and NO-dependent plasticity mechanisms to fit the parameters for the two pathways separately (see Materials and methods). In the resulting model, NO-dependent plasticity develops more gradually and requires more KC-DAN pairings to produce a memory of equal magnitude, compared to DA-dependent plasticity (*Figure 8C*).

We next asked how these plasticity mechanisms interact to determine effective KC-to-MBON synaptic weights when both DA and NO pathways are active. The synaptic weight is a function of both DA and NO-mediated effects, $w = f(D,N)$. One possibility for the function $f$ is a difference between terms corresponding to DA-dependent depression and NO-dependent facilitation; that is, $w \propto N - D$. When we fit a model with this functional form to our data, we found that it incorrectly predicts a reduction in memory strength after repeated pairings, because of the slower accumulation of NO-dependent facilitation after DA-dependent potentiation has saturated (*Figure 8D*, gray curve; *Figure 8C*). Another possibility for $f$ is a multiplicative form, for example $w \propto (1 - D) \times (1 + N)$. While we cannot unambiguously determine the identity of the biophysical parameters underlying DA and NO-mediated effects, such a form would arise naturally if the two pathways modulated parameters such as quantal size and the probability of synaptic vesicle release from KCs. We found that the multiplicative model provides a good match to our data (*Figure 8C*, blue curve). In this model, strong DA-dependent depression (i.e. $D$ close to 1) leads to a small synaptic weight even in the presence of NO-dependent facilitation.

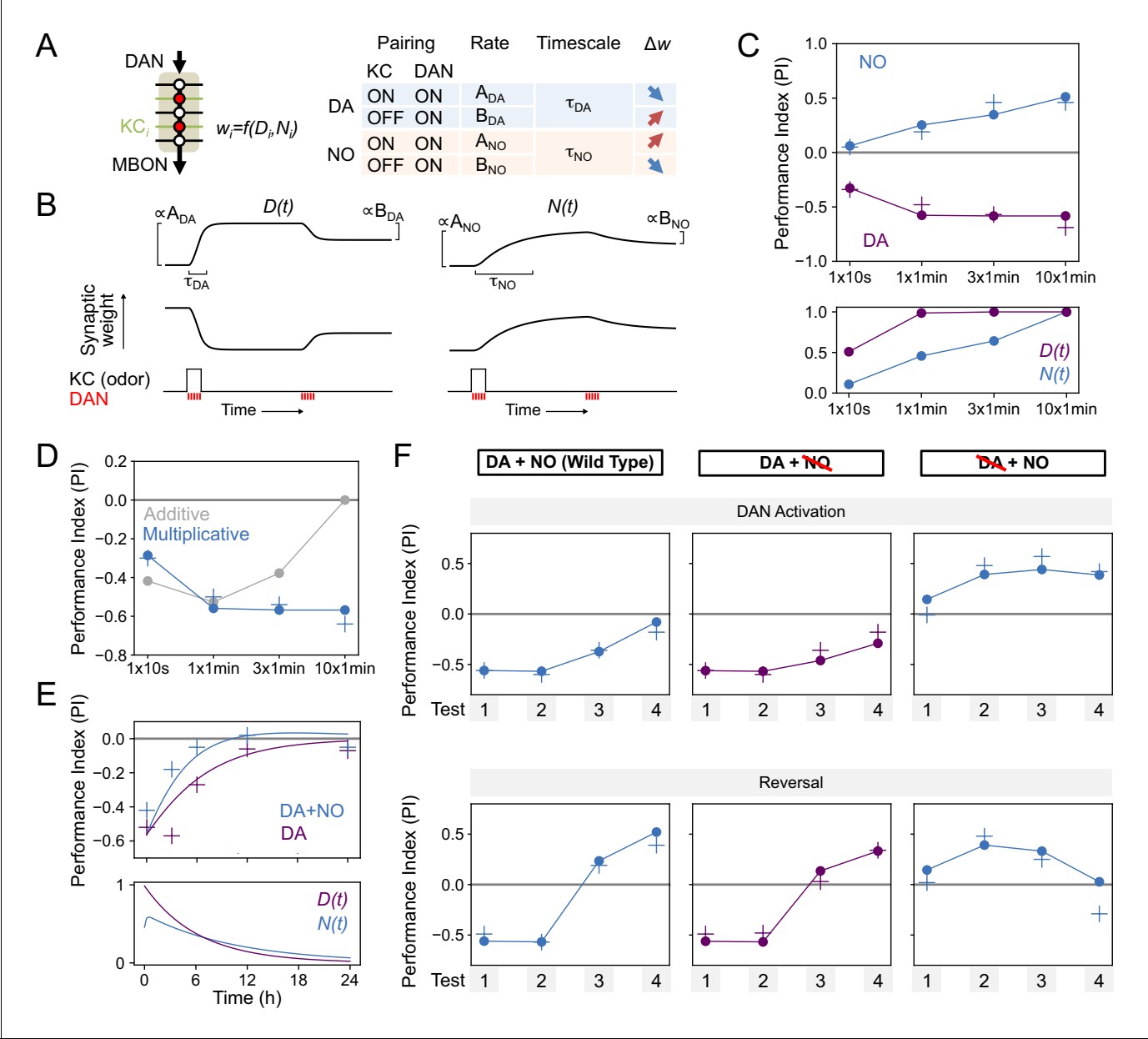

**Figure 8.** Modeling dopamine and NO mediated plasticity (A) Summary of plasticity model for independent dopaminergic (DA) and NO pathways. Synaptic weight $w_i$ from $KC_i$ to an MBON is increased or decreased depending on the pairing protocol. $A$ and $B$ determine the magnitude of the depression or potentiation induced by pairing, and $\tau$ determines the timescale over which weight changes occur. (B) Illustration of the effects of the model in (A), for only DA (left) or only NO (right) dependent plasticity. In each plot, KC and DANs are first co-activated, followed by a later DA activation without KC activation. (C) Top: Model performance index (PI) for different pairing protocols. In (C)-(F), crosses represent the means of data from *Figure 7*. Bottom: Dynamics of $D(t)$ and $N(t)$ in the model. (D) Modeling effects of combined DA and NO dependent plasticity. Gray curve: synaptic weights $w(t)$ are modeled as an additive function of DA and NO dependent effects $D(t)$ and $N(t)$, $w(t) \propto N(t) - D(t)$. Blue curve: a multiplicative interaction with $w(t) \propto (1 + N(t))(1 - D(t))$. (E) Modeling 24 hr memory decay following $1 \times 1$ min odor pairing. We assume a low level of spontaneous DAN activity and choose $B_{DA}$ and $B_{NO}$ to fit the data. Top: Performance index in model and data. Blue curve: control. Purple curve: only DA-dependent plasticity (compared to data from NOS-RNAi experiment). Bottom: Dynamics of $D(t)$ and $N(t)$ in the model. (F) Modeling effects of DAN activation and reversal learning. $B_{DA}$ and $B_{NO}$ are chosen to fit the effects of DAN activation (top). The model qualitatively reproduces the effects of reversal learning (bottom) with no free parameters.

The online version of this article includes the following figure supplement(s) for figure 8:

**Figure supplement 1.** Parameters for modeling dopamine and NO mediated plasticity.

**Figure supplement 2.** Performance of additive model on DAN activation and reversal paradigms.

We further tested our model by investigating its behavior in other paradigms. Assuming that spontaneous activity in DANs leads to memory decay (*Plaçais et al., 2012*; *Berry et al., 2018*; *Sitaraman et al., 2015*) accounted for the NO-dependent reduction in memory lifetime that we observed (*Figure 8E*). Fitting the magnitudes of DA-dependent facilitation and NO-dependent depression in the absence of KC activity using the data of *Figure 7D* also predicted the dynamics of reversal learning and its facilitation by NO with no additional free parameters (*Figure 8F*; *Figure 8—figure supplement 1*). On the other hand, the additive model failed to accurately predict these dynamics (*Figure 8—figure supplement 2*). In total, modeling a multiplicative interaction between DA- and NO-dependent plasticity accounts for the immediate effects of these pathways on odor memories that we observed experimentally. A notable exception is that this model cannot account for the enhanced persistence of memories after 10x training for DA or NO-null conditions (*Figure 6E*), suggesting a recruitment of additional consolidation mechanisms after spaced conditioning, as previously proposed (*Pagani et al., 2009*; *Tully et al., 1994*; *Pai et al., 2013*; *Scheunemann et al., 2018*; *Cervantes-Sandoval et al., 2013*; *Miyashita et al., 2018*; *Huang et al., 2012*; *Akalal et al., 2011*). Also, we found that spaced training with PPL-α3 can induce long-lasting memory but it requires at least 3 hr after training to develop (*Figure 9A*). This observation is consistent with the report that protein synthesis in MBON-α3 in the first 3 hr time window after odor-shock spaced training is required for LTM formation (*Pai et al., 2013*; *Wu et al., 2017*), but the current model does not account for the postsynaptic parameters.

## Discussion

Evidence from a wide range of organisms establishes that dopaminergic neurons often release a second neurotransmitter, but the role of such cotransmitters in diversifying neuronal signaling is much less clear. In rodents, subsets of dopaminergic neurons co-release glutamate or GABA (*Maher and Westbrook, 2008*; *Stuber et al., 2010*; *Sulzer et al., 1998*; *Tecuapetla et al., 2010*; *Tritsch et al., 2012*). In mice and *Drosophila*, single-cell expression profiling reveals expression of diverse neuropeptides in dopaminergic neurons (*Croset et al., 2018*; *Poulin et al., 2014*). EM connectome studies of the mushroom body in adult and larval *Drosophila* reveal the co-existence of small-clear-core and large-dense-core synaptic vesicles in individual terminals of dopaminergic neurons (*Eichler et al., 2017*; *Takemura et al., 2017*); moreover, the size of the observed large-dense-core vesicles differs between DAN cell types (*Takemura et al., 2017*).

We found that NOS, the enzyme that synthesizes NO, was located in the terminals of a subset of DAN cell types. NOS catalyzes the production of nitric oxide (NO) from L-arginine. *Drosophila* NOS is regulated by $Ca^{2+}$/calmodulin (*Regulski and Tully, 1995*), as is the neuronal isoform of NOS in the mammalian brain (*Abu-Soud and Stuehr, 1993*), raising the possibility that NO synthesis might be activity dependent. Furthermore, the localization of the NOS1 protein in the axonal terminals of DANs (*Figure 3D*) is consistent with NO serving as a cotransmitter. Our conclusion that NO acts as a neurotransmitter is supported by the observation that NO signaling requires the presence of a putative receptor, soluble guanylate cyclase, in the postsynaptic Kenyon cells. This role contrasts with the proposed cell-autonomous action of NOS in the ellipsoid body, in which NO appears to target proteins within the NOS-expressing ring neurons themselves, rather than conveying a signal to neighboring cells (*Kuntz et al., 2017*).

The valence-inversion phenotype we observed when PPL1-γ1pedc was optogenetically activated in a dopamine-deficient background can be most easily explained if NO induces synaptic potentiation between odor-activated KCs and their target MBONs. Our modeling work is consistent with this idea, but testing this idea and other possible mechanisms for NO action will require physiological experiments.

### Antagonistic functions of dopamine and nitric oxide

During olfactory learning, the concentration of $Ca^{2+}$ in KC axons represents olfactory information. The coincidence of a $Ca^{2+}$ rise in spiking KCs and activation of the G-protein-coupled Dop1R1 dopamine receptor increases adenylyl cyclase activity (*Abrams et al., 1998*; *Boto et al., 2014*; *Byrne et al., 1991*; *Kim et al., 2007*; *McGuire et al., 2003*; *Tomchik and Davis, 2009*). The resultant cAMP in turn activates protein kinase A (*Gervasi et al., 2010*; *Skoulakis et al., 1993*), a signaling cascade that is important for synaptic plasticity and memory formation throughout the animal

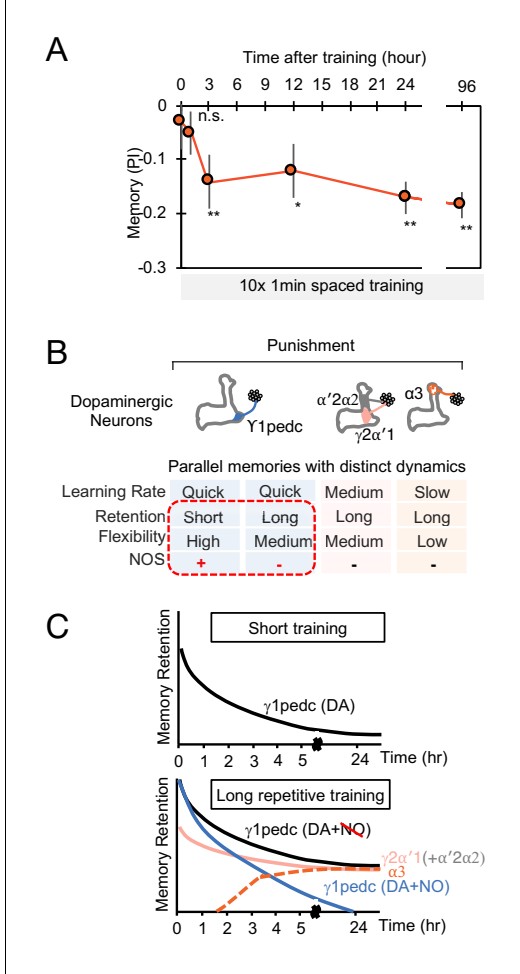

**Figure 9.** The MB stores parallel memories with distinct dynamics. (**A**) Flies were trained with 10 × 1 min spaced protocol with activation of PPL1-α3 (*Figure 6A*) and tested at different retention times. Memory was not detectable immediately after training and at 1 hr, but became significant after 3 hr. Data point and error bars indicate mean and SEM. N = 12–16. *, p<0.05; **, p<0.01 (**B**) Summary of this study. NOS diversify memory dynamics of MB compartments by reducing memory retention but enhancing flexibility. (**C**) A conceptual model of how naturalistic punishment (e.g. heat) induce memories that have distinct dynamics as flies experience training events. Punishment activate all PPL1-α3, γ2α′1 (MB-MV1) and γ1pedc (MB-MP1) to similar level (*Kirkhart and Scott, 2015*; *Mao and Davis, 2009*). PPL1-α′2/α′2 (MB-V1) does not respond to heat and shock (*Kirkhart and Scott, 2015*; *Mao and Davis, 2009*) and cannot induce aversive memory by itself, but can have a synergistic effect on memory retention when coactivated with PPL1-γ2α′1 (*Aso and Rubin, 2016*). Top: A one-time short training event first induces memory only in the γ1pedc compartment because of the fast learning rate by PPL1-γ1pedc (*Aso and Rubin, 2016*; *Hige et al., 2015*). This short-training is insufficient to induce NO-
*Figure 9 continued on next page*

phyla (e.g. *Davis et al., 1995*). In contrast, when DANs are activated without KC activity, and thus during low intracellular $Ca^{2+}$ in the KCs, molecular pathways involving the Dop1R2 receptor, Rac1 and Scribble facilitate decay of memory (*Berry et al., 2012*; *Cervantes-Sandoval et al., 2016*; *Kim et al., 2007*; *Shuai et al., 2010*).

We found that NOS in PPL1-γ1pedc shortens memory retention, while facilitating fast updating of memories in response to new experiences. These observations could be interpreted as indicating that NO regulates forgetting. Indeed, NO-dependent effect requires *scribble* in KCs, a gene previously reported as a component of active forgetting (*Figure 5D*). However, it is an open question whether the signaling pathways for forgetting, which presumably induce recovery from synaptic depression (*Berry et al., 2018*; *Cohn et al., 2015*), are related to signaling cascades downstream of NO and guanylate cyclase, which appear to be able to induce memory without prior induction of synaptic depression by dopamine. Lack of detectable 1-day memory formation after spaced training with PPL1-γ1pedc can be viewed as a balance between two distinct, parallel biochemical signals, one induced by dopamine and the other by NO (*Figure 6E*), rather than the loss of information (that is, forgetting). Confirming this interpretation will require better understanding of the signaling pathways downstream of dopamine and NO. The search for such pathways will be informed by the prediction from our modeling that dopamine and NO may alter two independent parameters that define synaptic weights with a multiplicative interaction.

In the vertebrate cerebellum, which has many architectural similarities to the MB (*Farris, 2011*; *Marr, 1969*; *Medina et al., 2002*), long-term-depression at parallel fiber-Purkinje cell synapses (equivalent to KC-MBON synapses) induced by climbing fibers (equivalent to DANs) can coexist with long-term-potentiation by NO (*Bredt et al., 1990*; *Lev-Ram et al., 2002*; *Shibuki and Okada, 1991*). In this case, the unaltered net synaptic weight results from a balance between coexisting LTD and LTP rather than recovery from LTD. This balance was suggested to play an important role in preventing memory saturation in the cerebellum and allowing reversal of motor learning. In the *Drosophila* MB, we observed a similar facilitation of reversal learning by NO (*Figure 7B*). The antagonistic roles of NO and synaptic depression may be a yet another common feature of the mushroom body and the cerebellum.

*Figure 9 continued*

dependent memory (*Figure 6B*). Bottom: When flies experience similar training repeatedly, stable memories will form in the compartments with slower learning rates (i.e. γ2α'1 and α3) (*Aso and Rubin, 2016*). Repetitive training also promotes NO-dependent processes to reduce memory retention and enhance flexibility in the γ1pedc. After repetitive training, PPL1-γ 1pedc also regulates stability of memories (*Awata et al., 2019*; *Plaçais et al., 2012*), but NO's role in that process is unknown.

## Distinct dynamics of dopamine and nitric oxide

Opposing cotransmitters have been observed widely in both invertebrate and vertebrate neurons (*Nusbaum et al., 2017*). A common feature in these cases is that the transmitters have distinct time courses of action. For instance, hypothalamic hypocretin-dynorphin neurons that are critical for sleep and arousal synthesize excitatory hypocretin and inhibitory dynorphin. When they are released together repeatedly, the distinct kinetics of their receptors result in an initial outward current, then little current, and then an inward current in the postsynaptic cells (*Li and van den Pol, 2006*). In line with these observations, we found that dopamine and NO show distinct temporal dynamics: NO-dependent memory requires repetitive training and takes longer to develop than dopamine-dependent memory. What molecular mechanisms underlie these differences? Activation of NOS may require stronger or more prolonged DAN activation than does dopamine release. Alternatively, efficient induction of the signaling cascade in the postsynaptic KCs might require repetitive waves of NO input. Direct measurements of release of dopamine and NO, and downstream signaling events by novel sensors will be needed to address these open questions (*Chen et al., 2014b*; *Eroglu et al., 2016*; *Patriarchi et al., 2018*; *Sun et al., 2018*; *Tang and Yasuda, 2017*).

## Toward subcellular functional mapping of memory genes

Decades of behavioral genetic studies have identified more than one hundred genes underlying olfactory conditioning in *Drosophila* (*Keene and Waddell, 2007*; *McGuire et al., 2005*; *Thurmond et al., 2019*; *Walkinshaw et al., 2015*). Mutant and targeted rescue studies have been used to map the function of many memory-related genes encoding synaptic or intracellular signaling proteins (for example, rutabaga, DopR1/dumb, DopR2/DAMB, PKA-C1/DC0, Synapsin, Bruchpilot, Orb2 and Rac1) to specific subsets of Kenyon cells (*Akalal et al., 2006*; *Berry et al., 2012*; *Blum et al., 2009*; *Gervasi et al., 2010*; *Han et al., 1996*; *Kim et al., 2007*; *Knapek et al., 2011*; *Krüttner et al., 2015*; *McGuire et al., 2003*; *Niewalda et al., 2015*; *Pavot et al., 2015*; *Qin et al., 2012*; *Shuai et al., 2010*; *Skoulakis et al., 1993*; *Trannoy et al., 2011*; *Vanover et al., 2019*; *Zars et al., 2000*). However, it is largely unknown if these proteins physically colocalize at the same KC synapses to form intracellular signaling cascades. Some of these proteins might preferentially localize to specific MB compartments. Alternatively, they may distribute uniformly along the axon of Kenyon cells, but be activated in only specific compartments. Our identification of cell-type-specific cotransmitters in DANs enabled us to begin to explore this question.

We used optogenetic activation of specific DANs to induce memory in specific MB compartments, while manipulating genes in specific types of KCs. This approach allowed us to map and characterize the function of memory-related genes at a subcellular level. For example, the Gycbeta100B gene, which encodes a subunit of sGC, has been identified as 'memory suppressor gene' that enhances memory retention when pan-neuronally knocked down (*Walkinshaw et al., 2015*), but the site of its action was unknown. Gycbeta100B appears to be broadly dispersed throughout KC axons, based on the observed distribution of a Gycbeta100B-EGFP fusion protein (*Figure 5—figure supplement 1*). Our experiments ectopically expressing NOS in PPL1-α3 DANs that do not normally signal with NO is most easily explained if sGC is available for activation in all MB compartments (*Figure 4D–E*, *Figure 4—figure supplement 1*).

What are the molecular pathways downstream to cGMP? How do dopamine and NO signaling pathways interact in regulation of KC synapses? Previous studies and RNA-Seq data suggest several points of possible crosstalk. In cultured KCs from cricket brains, cGMP-dependent protein kinase (PKG) mediates NO-induced augmentation of a $Ca^{2+}$ channel current (*Kosakai et al., 2015*). However, we failed to detect expression of either of the genes encoding *Drosophila* PKGs (foraging and Pkg21D) in KCs in our RNA-Seq studies (*Supplementary file 1*). On the other hand, cyclic nucleotide-gated channels and the cGMP-specific phosphodiesterase Pde9 are expressed in KCs.

Biochemical studies have shown that the activity of sGC is calcium dependent and that PKA can enhance the NO-induced activity of sGC by phosphorylating sGC; sGC isolated from flies mutant for adenylate cyclase, *rutabaga*, show lower activity than sGC from wild-type brains (*Morton et al., 2005*; *Shah and Hyde, 1995*), suggesting crosstalk between the cAMP and cGMP pathways.

### The benefits of parallel memory units with heterogeneous dynamics

All memory systems must contend with a tension between the strength and longevity of the memories they form. The formation of a strong immediate memory interferes with and shortens the lifetimes of previously formed memories, and reducing this interference requires a reduction in initial memory strength that can only be overcome through repeated exposure (*Amit and Fusi, 1994*). Theoretical studies have argued that this tension can be resolved by memory systems that exhibit a heterogeneity of timescales, balancing the need for both fast, labile memory and slow consolidation of reliable memories (*Fusi et al., 2005*; *Lahiri and Ganguli, 2013*; *Benna and Fusi, 2016*). The mechanisms responsible for this heterogeneity, and whether they arise from complex signaling within synapses themselves (*Benna and Fusi, 2016*), heterogeneity across brain areas (*Roxin and Fusi, 2013*), or both, have not been identified.

We found that NO acts antagonistically to dopamine and reduces memory retention (*Figure 6*) while facilitating the rapid updating of memory following a new experience (*Figure 7*). Viewed in isolation, the NO-dependent reduction in memory retention within a single compartment may seem disadvantageous, but in the presence of parallel learning pathways, this shortened retention may represent a key mechanism for the generation of multiple memory timescales that are crucial for effective learning. During shock conditioning, for example, multiple DANs respond to the aversive stimulus, including PPL1-γ1pedc, PPL1-γ2α′1, PPL1-α3 (*Mao and Davis, 2009*; *Riemensperger et al., 2005*). We have shown that optogenetic activation of these DAN cell types individually induces negative-valence olfactory memory with distinct learning rates (*Aso and Rubin, 2016*). The NOS-expressing PPL1-γ1pedc induces memory with the fastest learning rate in a wild-type background, and we show here that it induces an NO-dependent memory trace when dopamine synthesis is blocked, with a much slower learning rate and opposite valence. Robust and stable NO-dependent effects were only observed when training was repeated 10 times (*Figure 6E*). Under such repeated training, compartments with slower learning rates, such as α3, form memory traces in parallel to those formed in γ1pedc (*Pai et al., 2013*; *Séjourné et al., 2011*; *Wu et al., 2017*). Thus, flies may benefit from the fast and labile memory formed in γ1pedc without suffering the potential disadvantages of shortened memory retention, as long-term memories are formed in parallel in other compartments (*Figure 9B*). The *Drosophila* MB provides a tractable experimental system to study the mechanisms and benefits of diversifying learning rate, retention, and flexibility in parallel memory units, as well as exploring how the outputs from such units are integrated to drive behavior.

## Materials and methods

### Flies

*Drosophila* strains used in this study are listed in the *Supplementary file 2* KEY RESOURCES TABLE. Crosses listed in *Supplementary file 3* were kept on standard cornmeal food supplemented with retinal (0.2 mM all-trans-retinal prior to eclosion and then 0.4 mM) at 21°C at 60% relative humidity in the dark. Female flies were sorted on cold plates at least 1 d prior to the experiments, and 4–10 day old flies were used for experiments. Additional drugs were administered by feeding with retinal containing fly food mixed with drugs. The L-DOPA (D9628, Sigma) or L-NNA were mixed directly into melted fly food at final concentrations of 1 mg/ml or 1–100 mM, respectively. S-(−)-Carbidopa (C1335, Sigma) was dissolved in 1 ml of water at 10x the final concentration and mixed with 9 ml of melted fly food.

### Olfactory learning assay

Groups of approximately 20 female flies, 4–10 day post-eclosion, were trained and tested at 25°C at 50% relative humidity in the fully automated olfactory arena for optogenetics experiments (*Aso and Rubin, 2016*; *Pettersson, 1970*; *Vet et al., 1983*). The odors were diluted in paraffin oil (Sigma–Aldrich): 3-octanol (OCT; 1:1000; Merck) and 4-methylcyclohexanol (MCH; 1:750; Sigma–Aldrich).

Videography was performed at 30 frames per second and analyzed using Fiji (*Schindelin et al., 2012*). Statistical comparisons were performed (Prism; Graphpad Inc, La Jolla, CA 92037) using the Kruskal Wallis test followed by Dunn's post-test for multiple comparison, except those in *Figures 4C, E, 6B, C and E* and E, 9A, *Figure 4—figure supplement 1* and *Figure 5—figure supplement 3C*, which used Wilcoxon signed-rank test with Bonferroni correction to compare from zero. Appropriate sample size for olfactory learning experiment was estimated based on the standard deviation of performance index in previous study using the same assay (*Aso and Rubin, 2016*). We set the effect size, power and significance as 0.15, 0.8 and 0.05, respectively.

## Immunohistochemistry

Dissection and immunohistochemistry of fly brains were carried out as previously described with minor modifications (*Jenett et al., 2012*) using the antibodies listed in KEY RESOURCES TABLE. Brains and VNCs of 3- to 10-day-old female flies were dissected in Schneider's insect medium and fixed in 2% paraformaldehyde in Schneider's medium for 55 min at room temperature (RT). After washing in PBT (0.5% Triton X-100 in PBS), tissues were blocked in 5% normal goat serum (or normal donkey serum, depending on the secondary antibody) for 90 min. Subsequently, tissues were incubated in primary antibodies diluted in 5% serum in PBT for 2–4 days on a nutator at 4°C, washed four times in PBT for 15 min or longer, then incubated in secondary antibodies diluted in 5% serum in PBT for 2–4 days on a Nutator at 4°C. Tissues were washed thoroughly in PBT four times for 15 min or longer and mounted on glass slides with DPX.

For immunolabeling of NOS, the serum against NOS exon 16 was obtained from N. Yakuobovich and P. H. O'Farrell (*Yakubovich et al., 2010*), and then affinity purified as described below. In order to minimize non-specific signals, we absorbed 200 µL of anti-NOS antibody (1:1000) for 1 day with 30 fly brains in which NOS was knocked down panneuronally using elav-GAL4 and NOS-RNAi strain#1 (TRiP.HMC03076), and the supernatant was used for subsequent immunohistochemistry.

## Purification of dNOS proteins and antibody

The pRSET-dNOS exon 16 construct, containing an N-terminal His tag and T7 gene 10 leader RBS site, was assembled as follows. The NEBuilder Assembly tool was used to design primers for the NEBuilder HiFi DNA assembly (New England Biolabs # E2621S) of dNOS exon 16 as contained in pET28a-dNOS exon 16 (gift of Nikita Yakubovich, O-Farrell lab, UCSF) into the backbone vector pRSET (gift of Ariana Tkachuk, Janelia) which was digested with BamHI/EcoRI. The assembled product, pRSET-dNOS exon 16, was first transformed into NEB 5-alpha competent cells (New England Biolabs #E2621S) and plated on LB plus ampicillin (60 µl/ml).

For protein purification, pRSET-dNOS exon 16 DNA was then transformed into T7 Express lysY/lq *E. coli* protein expression cells (New England Biolabs #C3013) and plated on LB plus ampicillin (60 µl/ml). For large-scale growth, 500 ml of Miller's LB plus ampicillin (60 µl/ml) was inoculated from 5 ml of a starter culture and grown for ~3 hr (~O.D. 0.5–0.7) at 37°C and then induced by adding 0.5 mM IPTG. The culture was allowed to grow at 18–20°C overnight before spinning down and freezing the recovered pellets which were divided in two 250 ml bottles.

To resuspend the thawed pellets (frozen overnight), 10–12 ml of the nonionic detergent- based lysis reagent B-PER (Thermo Scientific #78266) in phosphate buffer containing 1 mg/ml of lysozyme (Thermo Scientific #89833), nuclease (0.1 µl/ml, Thermo Scientific # 88701), and 1X HALT protease inhibitors (Thermo Scientific #1861279). The suspension, divided into two 50 ml conical tubes, was gently shaken for 15–20 min at 30°C before spinning down at 8000 x g for 15 mins at 4°C. We found that that the majority of the dNOS exon 16 protein was in inclusion bodies and therefore we carried out purification starting with the pellet.

Each pellet was resuspended in ~10 ml B-PER containing 200 µg/ml lysozyme. The suspension was then mixed with 100 ml of a wash buffer containing 1:10 B-PER in ice cold 1xPBS (diluted from 10X PBS stock, Fisher Scientific #BP3994) by pipetting the mixture up and down and gentle agitation. The mixture was spun down at 15,000 rpm for 15 min at 4°C. The pellet was similarly washed four more times. The washed pellet was either stored at −20°C overnight or resuspended in 7–12 ml of inclusion body solubilization reagent (Thermo Scientific #78115). The protein suspension was shaken for 30–40 min at 20°C and then ultracentrifuged at 35,000 x g for 20–30 min at 4°C.

For affinity purification of anti-dNOS antibody, the supernatant fraction was concentrated using 50 ml conical tubes Vivaspin 20, 10,000 MWCO concentrators (Sartorius # VS2002) and then dialyzed in 3–12 ml dialysis cassettes 10,000 MWCO (Thermo Scientific #66810) against 1 L of 4 M guanidine HCL (diluted from 6 M stock, Sigma #SRE0066) in 1X PBS pH 8.0 for ~6 hr at 4˚C. The medium was further diluted to 2 M guanidine HCL and the protein continued to dialyze overnight. Affinity purification of the protein-antibody complex was performed using the AminoLink Immobilization Kit (Thermo Scientific #44890). Approximately 6 mg of soluble dNOS exon 16 protein in 2 M guanidine in 1X PBS was bound to the agarose beads in the column as antigen, and 1.9 ml of crude rabbit anti sera to dNOS exon 16 (gift of Nikita Yakubovich and Patrick O-Farrell) was run through the column. The purified dNOS exon 16 rabbit anti-antibody fractions were eluted with IgG Elution buffer (Thermo Scientific # 21004) and then concentrated with Vivaspin 20 tube concentrators before dialyzed in 1X PBS at 4˚C for 2.5 days with one change of fresh 1X PBS.

## Tissue expansion

Tissue expansion was performed as described in *Tillberg et al. (2016)*. All solutions were prepared in milliQ-grade water unless otherwise specified. AcX stock: acryloyl-X, SE (ThermoFisher, A20770) at 10 mg/mL in anhydrous DMSO. PLL solution: poly-l-lysine solution (Ted Pella 18026) with Photo-Flo detergent (EMS 74257) added 1:500. Acrylate stock: 4 M, prepared by neutralizing 5.5 mL acrylic acid (99% purity; Sigma, 147230) with 10 N NaOH using a water bath and fume hood, in a total volume of 20 mL. Acrylamide stock: 50% (w/v) (Sigma, A9099). Bisacrylamide stock: 1% (w/v) (Sigma, M7279). Monomer stock: 11.5 mL sodium acrylate stock, 2.5 mL acrylamide stock, 7.5 mL bisacrylamide stock, 18 mL 5 M NaCl (Sigma, S5150-1L), 5 mL 10xPBS (ThermoFisher, 70011044), and 2.5 mL water for a total volume of 47 mL. 4-HT stock: 4-hydroxy TEMPO at 0.5% (w/v) (Sigma, 176141). TEMED stock: N,N,N′,N′-Tetramethylethylenediamine at 10% (v/v) (Sigma, T7024). APS stock: ammonium persulfate at 10% (w/v) (Sigma, A3678). ProK digestion buffer: 0.5% Triton X-100, 500 mM NaCl, 1 mM EDTA, 50 mM Tris pH8. Appropriate caution was exercised when handling acrylamide, a known toxin.

Dissected, fixed, and antibody-stained *Drosophila* brains were treated with AcX stock solution diluted 1:100 in 1xPBS, with shaking, overnight. Brains were then washed with 1xPBS. A gelation chamber was created by applying a Press-to-Seal silicone gasket (ThermoFisher, P24740) to a glass slide, which was then coated with PLL solution. AcX-treated brains were immobilized on the PLL surface, up to nine per gasket. Gelation solution was created by adding 10 μL each of 4HT, TEMED, and APS stock solutions to 470 μL of monomer stock solution on ice. Brains were washed with gelation solution and then the gelation chamber was filled with ~200 μL of gelation solution and incubated on ice for 25 min. The gelation chamber was then sealed with a cover slip and placed in a 37˚C incubator to gel and cure for 2 hr.

Gelation chambers were disassembled and individual gels trimmed close to each brain. Gels were trimmed to a right trapezoid shape to facilitate keeping track of specimen orientation. Gels were incubated with proteinase K digestion enzyme (NEB, P8107S) diluted 1:100 in proK digestion buffer with shaking, overnight. Digested gels were washed with water 4 × 30 min followed by equilibration overnight.

## Image acquisition and analysis

We used a LSM710 confocal microscope (Zeiss; 20x/0.8 M27 or 63x/1.40 oil immersion objective) for imaging brains and a custom-made lattice light sheet microscope for imaging expanded brains as previously described (*Chen et al., 2014a*; *Gao et al., 2019*). Images were analyzed using Fiji (http://fiji.sc/), and visualized with VVD_Viewer (https://github.com/takashi310/VVD_Viewer/blob/master/README.md), a modified version of Fluorender (http://www.sci.utah.edu/software/13-software/127-fluorender.html; *Wan et al., 2012*).

## Fluorescence in situ hybridization (FISH)

FISH probe libraries were designed based on transcript sequences and were purchased from Biosearch Technologies. The FISH protocol and dye labeling procedures were described previously (*Long et al., 2017*). FISH probes for detecting tyrosine hydroxylase transcripts were described (*Meissner et al., 2019*). FISH probes for NOS are listed in KEY RESOURCES TABLE. Each probe

contains a 3'-end amine-modified nucleotide that allows direct coupling to an NHS-ester Cy3 dye (GE Healthcare, PA23001) according to the manufacturer's instructions. The brains of 3–5 day old adult flies were dissected in 1xPBS and fixed in 2% paraformaldehyde diluted PBS at room temperature for 55 min. Brain tissues were washed in 0.5% PBT, dehydrated, and stored in 100% ethanol at 4˚C. After exposure to 5% acetic acid at 4˚C for 5 min, the tissues were fixed in 2% paraformaldehyde in 1xPBS for 55 min at 25˚C. The tissues were then washed in $1 \times$ PBS with 1% of $NaBH_4$ at 4˚C for 30 min. Following a 2 hr incubation in prehybridization buffer (15% formamide, $2 \times$ SSC, 0.1% Triton X-100) at 50˚C, the brains were introduced to hybridization buffer (10% formamide, 2x SSC, 5x Denhardt's solution, 1 mg/ml yeast tRNA, 100 µg/ml, salmon sperm DNA, 0.1% SDS) containing FISH probes at 50˚C for 10 hr and then at 37˚C for an additional 10 hr. After a series of wash steps, the brains were dehydrated, cleared in xylene, and mounted in DPX. Image Z-stacks were collected using an LSM880 confocal microscope fitted with an LD LCI Plan-Apochromat 25x/0.8 oil or Plan-Apochromat 63x/1.4 oil objective after the tissue cured for 24 hr.

## Calcium imaging

Flies were reared at 25˚C on cornmeal medium supplemented with retinal (0.2 mM) that was shielded from light. Flies were transferred to cornmeal media mixed with 100 mM L-NG-Nitroargenine (L-NNA) 0.4 mM retinal while siblings used for comparison were transferred to the same media without L-NNA 24 hr before testing. All experiments were performed on female flies, 3–5 days posteclosion. Brains were dissected in a saline bath (103 mM NaCl, 3 mM KCl, 2 mM CaCl2, 4 mM MgCl2, 26 mM NaHCO3, 1 mM NaH2PO4, 8 mM trehalose, 10 mM glucose, 5 mM TES, bubbled with 95% O2/5% CO2). After dissection, the brain was positioned anterior side up on a coverslip in a Sylgard dish submerged in 2 ml saline at 20˚C.

The sample was imaged with a resonant scanning 2-photon microscope with near-infrared excitation (920 nm, Spectra-Physics, INSIGHT DS DUAL) and a 25X objective (Nikon MRD77225 25XW). The microscope was controlled by using ScanImage 2015.v3 (Vidrio Technologies)15. Volumes were acquired with 141 µm $\times$ 141 µm field of view at $512 \times 512$ pixel resolution at 2 µm steps over 42 slices, at approximately 1 Hz. The excitation power for Ca2+ imaging measurement was 12 mW.

For photostimulation, the light-gated ion channel CsChrimson was activated with a 660 nm LED (M660L3 Thorlabs) coupled to a digital micromirror device (Texas Instruments DLPC300 Light Crafter) and combined with the imaging path with a FF757-DiO1 dichroic (Semrock). On the emission side, the primary dichroic was Di02-R635 (Semrock), the detection arm dichroic was 565DCXR (Chroma), and the emission filters were FF03-525/50 and FF01-625/90 (Semrock). Photostimulation from PPL1-γ1pedc to MBON-γ1pedc occurred 5 times at 30 s intervals at maximum intensity of 1.2 mW/mm2 as show *Figure 5—figure supplement 1*. When testing the PPL1-γ1pedc to the PAM-DAN connection there were 9 stimulations at 30 s intervals with an intensity of 0.89 mW/mm2, as measured using Thorlabs S170C power sensor. The stimulation duration was 1 s in all trials.

After testing, laser power was increased to take two color high-resolution images containing fluorescence from both red and green channels. Using custom python scripts and features in the images, regions of interest (ROIs) were draw corresponding to the mushroom body compartments. Fluorescence in a background ROI, that contained no endogenous fluorescence, was subtracted from the mushroom body ROIs. In the ΔF/F calculations, baseline fluorescence is the median fluorescence over a 5-s time period before stimulation started. The ΔF is the fluorescence minus the baseline. Then the ΔF is divided by baseline to normalize the signal (ΔF/F). The final signal was run through a gaussian filter (sigma = 1). When more than one trial was performed, the mean ΔF/F was calculated over the trials and then the mean between animals.

## RNA-Seq

### Expression checks

Neurons of interest were isolated by expressing a fluorescent protein, either mCD8-GFP or tdTomato, using split-Gal4 drivers specific for particular cell types and then manually picking the fluorescent neurons from dissociated brain tissue. As a preliminary to the sorting process, each driver/reporter combination was 'expression checked' to determine if the marked cells were sufficiently bright to be sorted effectively and if there was any off-target expression in neurons other than those

of interest. Drivers that met both these requirements were used in sorting experiments as described below.

## Sorting of fluorescent-labeled neurons

*Drosophila* adults were collected daily as they eclosed, and aged 3–5 days prior to dissection. For each sample, 60–100 brains were dissected in freshly prepared, ice cold Adult Hemolymph Solution (AHS; 108 mM NaCl, 5 mM KCl, 2 mM CaCl$_2$, 8.2 mM MgCl$_2$, 4 mM NaHCO$_3$, 1 mM NaH$_2$PO$_4$, 5 mM HEPES, 6 mM Trehalose, 10 mM Sucrose), and the major tracheal branches removed. The brains were transferred to an 1.5 ml Eppendorf tube containing 500 microliters 1 mg/ml Liberase DH (Roche, prepared according to the manufacturer's recommendation) in AHS, and digested for 1 hr at room temperature. The Liberase solution was removed and the brains washed three times with ice cold AHS. The final wash was removed completely and 400 µl of AHS+2% Fetal Bovine Serum (FBS, Sigma) were added. The brain samples were gently triturated with a series of fire-polished, FBS-coated Pasteur pipettes of descending pore sizes until the tissue was homogenized, after which the tube was allowed to stand for 2–3 min so that the larger debris could settle.

For hand sorting, the cell suspension was transferred to a Sylgard-lined Glass Bottom Dish (Willco Wells), leaving the debris at the bottom of the Eppendorf tube, and distributed evenly in a rectangular area in the center of the plate with the pipet tip. The cells were allowed to settle for 10–30 min prior to picking. Fluorescent cells were picked with a mouth aspirator consisting of a 0.8 mm Nalgene Syringe Filter (Thermo), a short stretch of tubing, a plastic needle holder, and a pulled Kwik-Fil Borosilicate Glass capillary (Fisher). Cells picked off the primary plate were transferred to a Sylgard-lined 35 mm Mat Tek Glass Bottom Microwell Dishes (Mat Tek) filled with 170 microliters AHS+2% FBS, allowed to settle, and then re-picked. Three washes were performed in this way before the purified cells were picked and transferred into 50 µl buffer XB from the PicoPure RNA Isolation Kit (Life Technologies), lysed for 5 m at 42°C, and stored at −80°C.

For FACS sorting, the cell suspension was passed through a Falcon 5 ml round-bottom tube fitted with a 35 micrometer cell strainer cap (Fisher), and sorted on a Becton Dickson FACSAria II cell sorter, gated for single cells with a fluorescence intensity exceeding that of a non-fluorescent control. Positive events were sorted directly into 50 microliters PicoPure XB buffer, the sample lysed for 5 m at 42°C, and stored at −80°C.

## Library preparation and sequencing

Total RNA was extracted from 100 to 500 pooled cells using the PicoPure kit (Life Technologies) according to the manufacturer's recommendation, including the on-column DNAse step. The extracted RNA was converted to cDNA and amplified with the Ovation RNA-Seq System V2 (NuGEN), and the yield quantified by NanoDrop (Thermo). The cDNA was fragmented and the sequencing adaptors ligated onto the fragments using the Ovation Rapid Library System (NuGEN). Library quality and concentration was determined with the Kapa Illumina Library Quantification Kit (KK4854, Kapa Biosystems), and the libraries were pooled and sequenced on an Illumina NextSeq 550 with 75 base pair reads. Sequencing adapters were trimmed from the reads with Cutadapt (*Martin, 2011*) prior to alignment with the STARsolo aligner (https://github.com/alexdobin/STAR/blob/master/docs/STARsolo.md) (*Dobin et al., 2013*; *Dobin, 2020*) to the *Drosophila* r6.34 genome assembly on Flybase (*Thurmond et al., 2019*). The resulting transcript alignments were passed to RSEM (*Li and Dewey, 2011*) to generate gene expression counts. The data set was deposited to NCBI Gene Expression Omnibus (accession number GSE139889).

## Modeling

### Inferring parameters of DA and NO plasticity

We assume that the immediate effects of DA-mediated and NO-mediated changes at a KC-to-MBON synapse are described by two variables *d(t)* and *n(t)*, respectively. When the KC and corresponding DAN are coactive, these variables are modified according to:

$$\frac{d}{dt}d(t) = A_D(1 - d(t))$$

$$\frac{d}{dt}n(t) = A_N(1 - n(t)) \tag{1}$$

When the DAN is active but the KC is inactive,

$$\frac{d}{dt}d(t) = -B_D d(t)$$

$$\frac{d}{dt}n(t) = -B_N n(t) \tag{2}$$

A and B determine how quickly the variables approach their maximum value of 1 or minimum value of 0.

To model the time it takes for the effects of synaptic plasticity to occur, we also define two additional variables

$$\tau_D \frac{d}{dt}D(t) = d(t) - D(t)$$

$$\tau_N \frac{d}{dt}N(t) = n(t) - N(t) \tag{3}$$

$D(t)$ and $N(t)$ follow the values of $d(t)$ and $n(t)$, but with slower timescales $\tau_D$ and $\tau_N$ respectively. Based on the data, we assume $\tau_D$ = 30 s, and $\tau_N$ = 10 min. This accounts for the slower induction of NO-mediated effects.

We start by inferring AD and AN from data by relating the values of $D(t)$ and $N(t)$ in our model to the activation of the MBON, and finally to the PI. We assume that the normalized KC-to-MBON synaptic weight is given by $w(t) \propto 1 - D(t)$ (DA-mediated depression) if NO is absent, or $w(t) \propto 1 + N(t)$ (NO-dependent facilitation) if DA is absent. In our model, odors A and B activate a random 10% of KCs (the results do not depend on the total number of KCs $N_{KC}$), and the activation of the MBON is given by $r = \frac{1}{N_{KC}} \sum_i^f w_i s_i$, where $s_i = 1$ if the $i$th KC is active and 0 otherwise. At the beginning of each trial, $D(t) = N(t) = 0$, so $w_i = 1$. If $r_A$ and $r_B$ are the MBON activations for odors A and B, then we assume the probability of the fly choosing odor A is equal to a softmax function of this activation:

$$P_{odorA} = \frac{e^{g r_A}}{e^{g r_A} + e^{g r_B}} \tag{4}$$

We also infer $g$, which determines how strongly the MBON activation influences the decision, with g = 0 corresponding to random choices.

We infer $A_D$, $A_N$, and g separately for DA-null and NO-null conditions. To do so, we determine the values of the parameters that minimize the mean squared distance between model prediction and experimentally measured PIs for the 1 × 10 s, 1 × 1 min, 3 × 1 min, and 10 × 1 min protocols (*Figure 6B*). This is accomplished by simulating the model defined by Equations *Equation 1* to *Equation 3* and calculating the resulting preference index using Equation *Equation 4*. Optimal values for $A_D$, $A_N$, and g are found using a grid search. The optimization leads to $A_D$ = 4.3 min$^{-1}$ and $A_N$ = 0.96 min$^{-1}$ (the inferred values of g are similar for the two cases; 14.8 and 12.6, respectively). These values indicate that the DA effect saturates more quickly than the NO effect, nearly reaching its maximum effect after 1 × 1 min pairing.

Next, we model synaptic weights when both DA and NO-dependent changes occur. We consider two forms of interaction, an additive one with $w(t) \propto N(t) - D(t)$ and a multiplicative one with $w(t) \propto (1 - D(t))(1 + N(t))$. We use the values of $A_D$ and $A_N$ inferred previously but allow g to be readjusted to best match the data (for the multiplicative model, g = 14.3, similar to above, while for the additive model g = 24.6). Only the multiplicative model qualitatively matches the experimental data (*Figure 8D*).

## Modeling memory decay after 1 × 1 min pairing

Next, we ask how memory decays after a pairing protocol. We assume that, after the pairing protocol is complete, there is a background level of activity in the DANs, which leads to depression

according to Eq. 2. We infer the values of $B_D^{bg}$ and $B_N^{bg}$ (where the superscript denotes decay due to background DAN activity) using a grid search to minimize the mean squared distance between the predicted and actual PI from Figure 6D. Other parameters are set to the previously determined values using the multiplicative model above. This leads to $B_D^{bg}$ = 2.7 x$10^{-3}$ min$^{-1}$) and $B_D^{bg}$ = 1.6 x$10^{-3}$ min$^{-1}$).

## Predicting memory dynamics in reversal learning

Finally, we also model the effects of different pairing protocols. We start by considering DAN activation in the absence of odor. We infer values for $B_D$ and $B_N$ (different from the background levels above, since DANs are now activated rather than at their background levels of activation) by minimizing the mean squared difference between predicted and actual PI for NO-null and DA-null conditions (*Figure 7F*), and with the remaining parameters determined previously for the multiplicative model. This yields $B_D$ = 0.26 (min$^{-1}$) and $B_N$ = 0.16 (min$^{-1}$). These parameters are used to predict the behavior during reversal learning (*Figure 7F*).

## Acknowledgements

We thank Jui-Chun Kao, and the Janelia Fly Facility for help with fly husbandry. We thank Eizaburo Doi, Joshua Dudman, Yichun Shuai, Mehrab Modi, Karen Hibbard Gowan Tervo, Adam Hantman, Daisuke Hattori, Larry Abbott, Vanessa Ruta, Stott Waddell, Hiromu Tanimoto, Toshihide Hige, Brett Mensh and Glenn Turner for stimulating discussions and for comments on earlier drafts of the manuscript. Andrew Lin for fly stocks. Nikita Yakubovich and Patrick O'Farrell shared anti-NOS 16 serum pET28a-dNOS exon 16 plasmid. Phil Borden, Gaby Paez, Silvia Sanchez Martinez, Ariana Tkachuk, Tim Brown, Yan Zhang, Lauren Porter, Jonathan Marvin, and Loren Looger provided guidance on affinity purification of the anti-NOS antibody. Burkhard Poeck and Roland Strauß shared confocal images of fly brains stained with anti-NOS. Daniel E Milkie, Jennifer Colonell, Ruixuan Gao, Srigokul Upadhyayula, Eric Betzig, Michael DeSantis, Damien Alcor and Janelia Advanced Imaging Center for supporting the lattice light sheet imaging. Igor Pisarev, Stephan Saalfeld, Takashi Kawase and Hideo Otsuna provided help with image analysis. Rebecca Vorimo, Allison Sowell, Kari Close, Project Technical Resources, and the FlyLight Project Team at Janelia Research Campus provided brain dissection and histological preparation. The laboratory of JH was supported by NIH R01 GM GM84128.

## Additional information

### Funding

| Funder | Grant reference number | Author |
| --- | --- | --- |
| Howard Hughes Medical Institute | | Yoshinori Aso<br>Robert P Ray<br>Xi Long<br>Daniel Bushey<br>Teri-TB Ngo<br>Brandi Sharp<br>Christina Christoforou<br>Amy Hu<br>Andrew L Lemire<br>Paul Tillberg<br>Gerald M Rubin |
| National Institutes of Health | R01 GM84128 | Karol Cichewicz<br>Jay Hirsh |
| Simons Foundation | Global Brain | Yoshinori Aso<br>Ashok Litwin-Kumar<br>Gerald M Rubin |
| Burroughs Wellcome Fund | | Ashok Litwin-Kumar |
| Gatsby Charitable Foundation | | Ashok Litwin-Kumar |
| National Science Foundation | DBI-1707398 | Ashok Litwin-Kumar |

The funders had no role in study design, data collection and interpretation, or the decision to submit the work for publication.

## Author contributions

Yoshinori Aso, Conceptualization, Data curation, Formal analysis, Supervision, Validation, Investigation, Visualization, Methodology, Writing—original draft, Project administration, Writing—review and editing; Robert P Ray, Software, Formal analysis, Investigation, Methodology; Xi Long, Paul Tillberg, Investigation, Methodology; Daniel Bushey, Formal analysis, Investigation; Karol Cichewicz, Brandi Sharp, Jay Hirsh, Resources; Teri-TB Ngo, Resources, Investigation, Methodology; Christina Christoforou, Amy Hu, Investigation; Andrew L Lemire, Software, Formal analysis; Ashok Litwin-Kumar, Formal analysis, Writing—original draft, Writing—review and editing; Gerald M Rubin, Conceptualization, Funding acquisition, Writing—original draft, Project administration, Writing—review and editing

## Author ORCIDs

Yoshinori Aso (iD) https://orcid.org/0000-0002-2939-1688
Xi Long (iD) http://orcid.org/0000-0002-0268-8641
Daniel Bushey (iD) http://orcid.org/0000-0001-9258-6579
Ashok Litwin-Kumar (iD) http://orcid.org/0000-0003-2422-6576
Gerald M Rubin (iD) https://orcid.org/0000-0001-8762-8703

## Decision letter and Author response

Decision letter https://doi.org/10.7554/eLife.49257.sa1
Author response https://doi.org/10.7554/eLife.49257.sa2

## Additional files

### Supplementary files

• Supplementary file 1. RNA-seq data. Transcripts Per Kilobase Million (TPM) TPM values for each splicing isoform are listed with the corresponding FlyBase ID. The source data is available at NCBI Gene Expression Omnibus (accession number GSE139889).

• Supplementary file 2. Key resource table. The list of key reagents used in this study.

• Supplementary file 3. *Drosophila* genotypes. The list of *Drosophila* genotypes and drug treatment used in each experiment.

• Transparent reporting form

### Data availability

Complete transcript data were deposited to NCBI Gene Expression Omnibus (accession number GSE139889).

The following dataset was generated:

| Author(s) | Year | Dataset title | Dataset URL | Database and Identifier |
|---|---|---|---|---|
| Aso Y, Ray RP, Aswath K, Ballard S, Lemire A, Rubin GM | 2019 | Bulk RNA-seq data from the dopaminergic neurons, MB output neurons, Kenyon cells, APL and DPM neurons in adult Drosophila | https://www.ncbi.nlm.nih.gov/geo/query/acc.cgi?acc=GSE139889 | NCBI Gene Expression Omnibus, GSE139889 |

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
