## [Decision Letter]

**Acceptance summary:**

Previous work in *Drosophila* has shown that apparently equivalent dopaminergic neurons can act on the same postsynaptic odor-coding cells to encode memories that differ in terms of valence and duration as well as learning rules required for their induction. These discoveries have important implications for diverse functions and effects of dopaminergic modulation in the mammalian brain. In this paper, Aso, Rubin and colleagues further explore cellular mechanisms that confer different properties to modulation mediated by different types of dopaminergic neurons. The key general insight is dopaminergic neurons can differ in cotransmitters released and that the difference in rules of timescales of learning and memory supported by dopamine can be ascribed to the effect of cotransmitter released. This broad idea is specifically documented here for dopaminergic neurons that co-release the gaseous transmitter nitric oxide. Beautifully precise experiments show that nitric oxide and dopamine co-released from certain dopaminergic neurons respectively write memories of opposite valence with different timescales. While dopamine induces plasticity relatively rapidly, nitric oxide induces an opposite form of plasticity over a slightly slower time scale. In this manner, the action of the gaseous transmitter nitric oxide not only reduces memory retention in wild type flies, but also fosters faster memory updating (e.g. in reversal learning). Thus, the work: (a) directly explains differences in memory encoding between specific distinct dopaminergic neurons; (b) provides new information on nitric oxide function in the brain; and (c) provides a potential general model in which cotransmitters confer unique modulatory properties to otherwise equivalent dopaminergic neurons.

**Decision letter after peer review:**

Thank you for submitting your article "Nitric oxide acts as a cotransmitter in a subset of dopaminergic neurons to diversify memory dynamics" for consideration by *eLife*. Your article has been reviewed by two peer reviewers, and the evaluation has been overseen by Mani Ramaswami as Reviewing Editor and K VijayRaghavan as the Senior Editor. The following individual involved in review of your submission has agreed to reveal their identity: Roland H. Strauss (Reviewer #2).

The reviewers have discussed the reviews with one another and the Reviewing Editor has drafted this decision to help you prepare a revised submission.

Summary:

This is a wonderful study that demonstrates how in different compartments of the mushroom body, individual Kenyon cells can associate different valences with an odor using different learning rules with different timing. It also importantly demonstrates how different subtypes of superficially equivalent dopaminergic neurons can both encode different types of reinforcement and induce memories of different duration. The specific finding is that nitric oxide and dopamine co-released from certain dopaminergic neurons respectively write memories of opposite valence with different timescales. This finding is very well supported by behavioral studies combined with highly precise labeling and manipulation of dopaminergic subtypes as well as cell-type specific RNA-Seq analyses. Action of the gaseous transmitted nitric oxide not only reduces memory retention in wild type flies, but also fosters faster memory updating (e.g. in reversal learning). Using a computational model, the authors propose that dopamine and nitric oxide's effects on Kenyon cell output synapses might combine multiplicatively, not additively. This work demonstrates a diversity of dopaminergic neuron function and mechanisms, provides new insight into NO function in the brain and further extends the increasing parallels between the insect mushroom body and vertebrate cerebellum.

Essential revisions:

1) It may be worth acknowledging that the work (understandably) stops short of demonstrating that NO itself is the key messenger. Given the rigor of the rest of the work, perhaps it is worth considering a formal possibility that NOS1 signals to KCs via sGC in flies via a messenger different from NO?

2) The authors note in the legend to Figure 5 that the reduced inverted memory in MB-switch>Gycβ100B-RNAi flies without RU486 is probably due to leaky expression. A similar effect is seen in Figure 1 of Ferris et al. (Nat. Neurosci., 2006) and Figure 1 of Liu et al. (*eLife,* 2016). It is true that in previous papers, MB-switch without RU486 did not induce expression of GFP (Liu et al.) or lacZ (Mao et al., 2004 – original description of MB-switch). But maybe it depends on the UAS construct (e.g., perhaps GFP fluorescence is more sensitive than X-gal staining) or even the fly food. The authors may consider imaging some MB-switch>GFP brains with and without RU486. This should be very easy, would benefit the field by clarifying if MB-switch is leaky, and could support the authors' explanation for reduced inverted memory in the no-RU486 control for MB-switch>Gycβ100B-RNAi.

3) NO diffuses through membranes, so how is compartmentation kept up? Are NO-sensitive compartments interspaced with NO-insensitive compartments? Is a spill-over eventually meaningful? After all, "sGC [is] present in all MB compartments". The authors should discuss these conceptual issues given that NO is a diffusible gas. How is NO prevented from simply diffusing from the g1 compartment (NOS-positive) to the g2 compartment (NOS-negative) next door?

4) Is it surprising that TH-Gal4>shi[ts] does not cause inverted memory like the TH-null mutant (since NO signaling shouldn't require vesicle release)? Would the authors predict that TH-Gal4>shi[ts] would *not* block inverted memory in a TH-null mutant?

5) The narrative leading up to the discovery of NO as cotransmitter is probably historically accurate. However, given that the cotransmitter eventually discovered is a gas that isn't packaged into vesicles at all, the lead up describing how the presence of both clear and dense-core vesicles argues for a cotransmitter could be modified in a way that better prepares the reader for the final discoveries.

6) Subsection “Soluble guanylate cyclase in Kenyon cells is required to form NO-dependent memories”, last paragraph: The authors find that L-DOPA feeding in adults restores "normal" memory in TH mutants with MB>Gycβ100B-RNAi, and use this to argue that the lack of inverted memory in TH-null + MB>Gycβ100B-RNAi flies is not a developmental effect. Perhaps what the authors meant to write is that this control (like Figure 4A) shows that Gycβ100B-RNAi flies don't have a *general* defect in learning (as opposed to a specific defect in inverted learning)?

7) The model is very nice but the reader could use a bit more hand-holding in developing an intuition behind the results. Why exactly is it that the additive model "incorrectly predicts a reduction in memory strength after repeated pairings"? The stated explanation, "because of the slower accumulation of NO-dependent facilitation after DA-dependent potentiation has saturated" makes sense but doesn't give a very clear picture. Would it help to create a figure panel illustrating the time courses of *D* and *N* in the additive and multiplicative models and how these combine to affect *w*? Why (both practically and conceptually) isn't it possible to change the parameters for the additive model to allow the model to match the data in Figure 8D? The authors explain in the Materials and methods that the models were fit to the DA-only and NO-only data in Figure 6B, but the Results section would be easier to follow if this was also stated in the Results.

8) Does the additive model also fail to match experimentally measured memory decay/reversal? If so, that would help convince the reader that the close match between the data and model in Figure 8E-F isn't just an artefact from overfitting.

9) In reference to the second paragraph of the Discussion: Cell-autonomous NO signalling is also found in cerebellum Purkinje cells. Given the evidence for cell autonomy in two other systems, it could be useful to perform one additional experiment to test whether sGC is also required in DAN neurons. While appreciating that negative observations are hard to definitively interpret, this seems to be an issue worth addressing experimentally. If not, then the fact that dual targets for NO have not been excluded should be acknowledged. (This is also pertinent to comments 1 and 3).

---

## [Author Response]

Suggested revisions:1) It may be worth acknowledging that the work (understandably) stops short of demonstrating that NO itself is the key messenger. Given the rigor of the rest of the work, perhaps it is worth considering a formal possibility that NOS1 signals to KCs via sGC in flies via a messenger different from NO?

We have indeed detected expression of sGCs in dopaminergic neurons, which raises the possibility that NO might first act on sGC in dopaminergic neurons, which in turn would promote release of another cotransmitter that can activate sGC in KCs. While the neuropeptide, Nplp1 is expressed in PPL1-γ1pedc, the receptor of Nplp1 is not expressed in Kenyon cells arguing against the relevance of this signaling pathway. Nevertheless, we modified the text to explicitly state that “both MBONs and DANs express sGC (Figure 5B)”.

2) The authors note in the legend to Figure 5 that the reduced inverted memory in MB-switch>Gycβ100B-RNAi flies without RU486 is probably due to leaky expression. A similar effect is seen in Figure 1 of Ferris et al. (Nat Neurosci, 2006) and Figure 1 of Liu et al. (eLife, 2016). It is true that in previous papers, MB-switch without RU486 did not induce expression of GFP (Liu et al) or lacZ (Mao et al., 2004 – original description of MB-switch). But maybe it depends on the UAS construct (e.g., perhaps GFP fluorescence is more sensitive than X-gal staining) or even the fly food. The authors may consider imaging some MB-switch>GFP brains with and without RU486. This should be very easy, would benefit the field by clarifying if MB-switch is leaky, and could support the authors' explanation for reduced inverted memory in the no-RU486 control for MB-switch> Gycβ100B-RNAi.

We performed the requested experiment and observed leaky expression of GFP in

Kenyon cells without RU486 under our experimental conditions with two reporter lines

(Figure 5—figure supplement 1E and F). We also analyzed the effect of MB-switch>Gycβ100B-RNAi without RU486, as assessed using anti-GFP IHC of the Gycbeta100B-EGFP MiMIC line and found significant reduction of Gycβ100B-EGFP without feeding of RU486 (Figure 5—figure supplement 1C), an observation most easily explained by leaky expression of MB-switch.

3) NO diffuses through membranes, so how is compartmentation kept up? Are NO-sensitive compartments interspaced with NO-insensitive compartments? Is a spill-over eventually meaningful? After all, "sGC [is] present in all MB compartments". The authors should discuss these conceptual issues given that NO is a diffusible gas. How is NO prevented from simply diffusing from the g1 compartment (NOS-positive) to the g2 compartment (NOS-negative) next door?

It is correct that plasma membranes would not prevent diffusion, but NO is a highly reactive molecule (having a half-life of a few seconds) and is converted to nitrates and nitrites by oxygen and water, providing a sink that limits diffusion. NOS1-positive compartments are confined to the γ lobe and are separated by the NOS-negative γ2 compartment. We note that unlike other MB compartments that lie in close proximity to one another, γ1 and γ2 are separated by a ~10 µm gap (see Figure 10G and H in Aso et al., 2014a); such a gap might serve to limit cross talk between compartments.

The limited diffusion of NO is further supported by the fact that the effects of NOS1 depend on the identity of the NOS-expressing DAN. Activation of NOS1 in γ1 induced appetitive memory, whereas activation of NOS1 in γ5 induced aversive memory. This result is not consistent with long-range NO diffusion. We also added calcium imaging experiments (Figure 5—figure supplement 2) showing that activation of PPL1-γ1pedc in both TH wild type and mutant background can increase calcium levels in MBON-γ1pedc in a NOS-dependent manner. However, activation of PPL1-γ1pedc failed to increase calcium levels in PAM-γ3, γ4 and γ5, although these neurons express sGC (based on our RNA-Seq data), an observation most easily explained by NO being unable to diffuse to these distant compartments.

We do believe that all the MB compartments have the capacity to respond to NO, given that the receptor of NO (i.e. soluble guanylyl cyclase) is ubiquitous in KC axons (Figure 5—figure supplement 1). In support of this view, we added data that ectopic NOS expression in PPL1-γ2α’1 can generate a valence inverted memory phenotype (Figure 4—figure supplement 1).

4) Is it surprising that TH-Gal4>shi[ts] does not cause inverted memory like the TH-null mutant (since NO signaling shouldn't require vesicle release)? Would the authors predict that TH-Gal4>shi[ts] would not block inverted memory in a TH-null mutant?

Given that dynamin is dispensable for NOS activation, blocking the recycling of synaptic vesicle by shibire would not be expected to affect NO release, but would be expected to block dopamine release. In such a situation, NO would indeed be expected to induce a positive-valence memory in the γ1pedc compartment after odor-shock conditioning. However, as pointed out, blocking PPL1-DANs with TH-GAL4>shi resulted in only a reduced aversive memory – but not inversion of valence – after 1x 1-min odor-shock conditioning, as reported in previous studies (e.g. Schwaerzel et al., 2003; Aso et al., 2012). There are several possible explanations for this apparent discrepancy, which cannot be distinguished by our data. It is most easily explained by assuming that the shibire block of DA release was incomplete. We also note that electric shock activates PPL1-γ2α’1 that can itself induce an aversive memory even in the absence of dopamine, presumably mediated by a cotransmitter. If the release of this cotransmitter was not blocked by shibire, an aversive memory induced by electric shock in γ2α’1 might counteract the positive-valence memory in γ1pedc.

5) The narrative leading up to the discovery of NO as cotransmitter is probably historically accurate. However, given that the cotransmitter eventually discovered is a gas that isn't packaged into vesicles at all, the lead up describing how the presence of both clear and dense-core vesicles argues for a cotransmitter could be modified in a way that better prepares the reader for the final discoveries.

We would like to keep this historical narrative, as it accurately reflects how the work developed. Although we eventually attributed valence-inversion phenotype to NO, our RNA-Seq data also provides a list of neuropeptides that may be packaged into dense core vesicles and function as cotransmitters.

6) Subsection “Soluble guanylate cyclase in Kenyon cells is required to form NO-dependent memories”, last paragraph: The authors find that L-DOPA feeding in adults restores "normal" memory in TH mutants with MB>Gycβ100B-RNAi, and use this to argue that the lack of inverted memory in TH-null + MB>Gycβ100B-RNAi flies is not a developmental effect. Perhaps what the authors meant to write is that this control (like Figure 4A) shows that Gycβ100B-RNAi flies don't have a general defect in learning (as opposed to a specific defect in inverted learning)?

We have revised the text as follows:

“Although MB-switch showed significant leaky expression without RU-486 feeding under our experimental conditions (Figure 5—figure supplement 1E and F), we could restore normal aversive memory by feeding TH mutant flies L-DOPA (Figure 5C), indicating that flies carrying MB-switch driven Gycβ100B-RNAi do not have a general defect in learning.”

7) The model is very nice but the reader could use a bit more hand-holding in developing an intuition behind the results. Why exactly is it that the additive model "incorrectly predicts a reduction in memory strength after repeated pairings"? The stated explanation, "because of the slower accumulation of NO-dependent facilitation after DA-dependent potentiation has saturated" makes sense but doesn't give a very clear picture. Would it help to create a figure panel illustrating the time courses of D and N in the additive and multiplicative models and how these combine to affect w?

We agree that the addition of figures to track the changes in model parameters would aid understanding. We added separate plots for *D(t)* and *N(t)* in Figure 8C-E and Figure 8—figure supplement 1 and 2. This allows the reader to track the components of the model for each of the paradigms we tested.

Why (both practically and conceptually) isn't it possible to change the parameters for the additive model to allow the model to match the data in Figure 8D? The authors explain in the Materials and methods that the models were fit to the DA-only and NO-only data in Figure 6B, but the Results section would be easier to follow if this was also stated in the Results.

We have added the additive model in Figure 8—figure supplement 2. The additive model can also qualitatively replicate behavioral results, but the fit is worse compared to the multiplicative model. As in Figure 8D, the additive model fails to fit with behavioral results when more training is applied.

We revised the text as follows:.

“Figure 7D also predicted the dynamics of reversal learning and its facilitation by NO with no additional free parameters (Figure 8F; Figure 8—figure supplement 1). On the other hand, the additive model failed to accurately predict these dynamics (Figure 8—figure supplement 2).”

Additionally, we mention that fitting was done for the two pathways separately:

“We used data that isolates the effects of DA and NO dependent plasticity mechanisms to fit the parameters for the two pathways separately (see Materials and methods).”

8) Does the additive model also fail to match experimentally measured memory decay/reversal? If so, that would help convince the reader that the close match between the data and model in Figure 8E-F isn't just an artefact from overfitting.

The additive model only fits memory decay when repetition of training is low. Otherwise, as number of training repetition increases, the immediate memory score goes down and deviate from behavioral observation. In Figure 8—figure supplement 2, we show the behavior of the additive model on reversal learning, which exhibits a worse fit to the data.

9) In reference to the second paragraph of the Discussion: Cell-autonomous NO signalling is also found in cerebellum Purkinje cells. Given the evidence for cell autonomy in two other systems, it could be useful to perform one additional experiment to test whether sGC is also required in DAN neurons. While appreciating that negative observations are hard to definitively interpret, this seems to be an issue worth addressing experimentally. If not, then the fact that dual targets for NO have not been excluded should be acknowledged. (This is also pertinent to comments 1 and 3).

We are unable to conduct the suggested experiments in the allowed time period for technical reasons, but now state in the text that “our results do not exclude the possibility that NO has other targets. Indeed, both MBONs and DANs express sGC (Figure 5B)” thereby explicitly raising the possibility that NO also acts autonomously inside the DANs.

We also added some additional data in response to comments and suggestions we received from non-reviewer colleagues on the submitted version of our paper that was posted on bioRxiv:

1) We provided additional data on the locations of NOS expression by: (a) adding RNA-Seq results in Figure 3—figure supplement 2 on five additional cell types: two additional MBONs, KC-a'/b', MB-DPM and MB-APL and (b) including expansion microscopy data on NOS-antibody staining in the γ1 compartment (Figure 3F and G and Video 1).

2) We added an experiment showing the effect of RNAi directed against *scribble*, a gene that has been shown by others to be involved in forgetting (Figure 5D).

3} An experiment was added showing that NOS1 was dispensable for relief learning (Figure 5—figure supplement 3).

4) A description of the memory dynamics experimentally observed in the a3 compartment is now shown.